# Extending on-surface synthesis from 2D to 3D by cycloaddition with C$_{60}$

Pengcheng Ding[1], Shaoshan Wang[1], Cristina Mattioli[2], Zhuo Li[1], Guoqiang Shi[1], Ye Sun [3], André Gourdon [2], Lev Kantorovich [4], Flemming Besenbacher [5], Federico Rosei[6] & Miao Yu [1] ✉

As an efficient molecular engineering approach, on-surface synthesis (OSS) defines a special opportunity to investigate intermolecular coupling at the sub-molecular level and has delivered many appealing polymers. So far, all OSS is based on the lateral covalent bonding of molecular precursors within a single molecular layer; extending OSS from two to three dimensions is yet to be realized. Herein, we address this challenge by cycloaddition between C$_{60}$ and an aromatic compound. The C$_{60}$ layer is assembled on the well-defined molecular network, allowing appropriate molecular orbital hybridization. Upon thermal activation, covalent coupling perpendicular to the surface via [4 + 2] cycloaddition between C$_{60}$ and the phenyl ring of the molecule is realized; the resultant adduct shows frozen orientation and distinct sub-molecular feature at room temperature and further enables lateral covalent bonding via [2 + 2] cycloaddition. This work unlocks an unconventional route for bottom-up precise synthesis of three-dimensional covalently-bonded organic architectures/devices on surfaces.

Constructing heterostructured molecular layers with atomic/molecular level precision and extending molecular engineering from two to three dimensions hold great importance for both fundamental science and application purposes in the fields of nanoelectronic devices, photonics, and quantum science[1–5]. Although it is straightforward to pile up molecular layers by van der Waals (vdW) interaction or hydrogen bonding[1–5], assemblies based on noncovalent bonding often have lower thermal stability, conductivity, and mechanical properties compared with the covalently-bonded products[6–8]. As an efficient bottom-up approach, on-surface synthesis (OSS) defines a special opportunity to investigate intermolecular coupling at the sub-molecular level and has successfully delivered various covalently-bonded two-dimensional (2D) polymers[9–15]. So far, these systems have been all based on the lateral coupling of molecular precursors within a single molecular layer. Extending OSS from two to three dimensions requires not only the lateral coupling parallel to the

substrate but also the covalent bonding perpendicular to the substrate, e.g., interlayer coupling between molecular layers, which is yet to be realized.

As the most representative fullerene, C$_{60}$ and its derivatives have shown magnificent physical properties/potentials (e.g. pressure resistance of solid C$_{60}$[16], optical restriction of C$_{60}$ solution[17], superconductivity of alkali-metal doped C$_{60}$ crystal[18], magnetism of TDAE-C$_{60}$ compound[19]); the chemical reactions of C$_{60}$ with organic compounds have also sparked considerable interests[20–22]. Especially, the unique 3D conjugated electronic structure and the geometric combination of pentagons and hexagons of C$_{60}$ molecules make their reactions significantly distinct from those of graphene or planar polyaromatic molecules[23,24]. The electron-deficient curved π-conjugation allows C$_{60}$ to form adducts with various dienes (e.g., cyclopentadiene, furan, anthracene) as a dienophile;[23,25] due to the multiple non-planar reactive sites ([6,6] bonds), C$_{60}$ is expected to act

[1]School of Chemistry and Chemical Engineering, Harbin Institute of Technology, Harbin 150001, China. [2]CEMES-CNRS, Toulouse 31055, France. [3]School of Instrumentation Science and Engineering, Harbin Institute of Technology, Harbin 150001, China. [4]Department of Physics, King's College London, The Strand, London WC2R 2LS, UK. [5]Interdisciplinary Nanoscience Center (iNANO), Aarhus University, Aarhus 8000, Denmark. [6]INRS Centre for Energy, Materials and Telecommunications, Varennes J3X 1P7, Canada. ✉e-mail: miaoyu_che@hit.edu.cn

as a steric spherical "multi-plug" reactant[20,26,27]. However, the close-packed arrangement of $C_{60}$ molecules[28–32] and their strong interaction with the substrates[33–35] make the on-surface reaction of $C_{60}$ with other molecules rather difficult. Moreover, the coupling between $C_{60}$ and the targeting functional group(s) requires a specific spatial geometry[25,36,37]. There is no precedent for the on-surface reaction of $C_{60}$ with aromatic compounds yet.

Herein, we report the cycloaddition of $C_{60}$ and 3,5-bis(carboxylic acid)-phenyl-3-maleimide (BCPM, $C_{12}H_7NO_6$) and their adducts on Au(111) surface (Fig. 1). To electronically decouple $C_{60}$ from the substrate and especially provide an appropriate steric configuration for the coupling between the lowest unoccupied molecular orbital (LUMO) of $C_{60}$ and the highest occupied molecular orbital (HOMO) of BCPM, 2D ordered BCPM layer is constructed on Au(111) beneath $C_{60}$. As demonstrated by scanning tunneling microscopy (STM) imaging and density functional theory (DFT) calculations, interlayer [4 + 2] cycloaddition between $C_{60}$ and BCPM perpendicular to the substrate is triggered upon thermal activation. The resultant adduct, i.e., $C_{60}$-BCPM, shows obvious differences from pristine $C_{60}$, including the frozen orientation hence triangular sub-molecular feature observed at room temperature (RT), increased adsorption height, and distinct domain structure and reactivity. Especially, forming macrocycles with the hexamer rings with $C_{6v}$ symmetry dominated, lateral inter-molecular [2 + 2] cycloaddition among $C_{60}$-BCPMs parallel to the substrate is evidenced. Using $C_{60}$ and BCPM as the first example, this work presents an unconventional strategy for bottom-up synthesis of 3D covalently-bonded organic architectures, extending OSS from 2D to 3D by cycloaddition reactions with fullerenes.

## Results

### Self-assembled network of BCPM on Au(111)

Extended honeycomb network of BCPM (Fig. 2a and Supplementary Fig. 1) self-assembles on Au(111) at 390 K. The network is tiled by flower-like units composed of six molecules, with $|\mathbf{a}| = |\mathbf{b}| = 27.2 \pm 0.2$ Å and an angle of 60° between the two vectors. Although the lateral molecular arrangement is similar to that formed on graphene epitaxially grown on Cu(111) (G-Cu)[38], close inspection (Fig. 2b) reveals a distinct morphology: even though the bis(carboxylic acid)-phenyl group of each molecule is imaged to be a rounded, inverted triangle in both cases, the maleimide ring on Au(111) shows three sub-protrusions (two of them being brighter than the third) instead of a circular bell as imaged on G-Cu. This morphology of BCPM on Au(111) resembles that on bare Cu(111)[39].

Based on DFT calculations, the bis(carboxylic acid)-phenyl group adsorbs in a flat geometry at a height of 3.4 Å above Au(111), while maleimide adopts a tilted configuration with the ring rotated by 20.5° from the horizon (Fig. 2c). The network is primarily stabilized by i) intermolecular double O–H···O hydrogen bonding within the flower-like unit and ii) ligand bonding between the oxygen atom closer to the Au substrate in maleimide and Au atom enabled by the tilted maleimide ring hence the reduced O–Au distance (Fig. 2d, e). DFT-optimized structural model (Fig. 2d) and its calculated image (Fig. 2b) are in good agreement with the experimental observations.

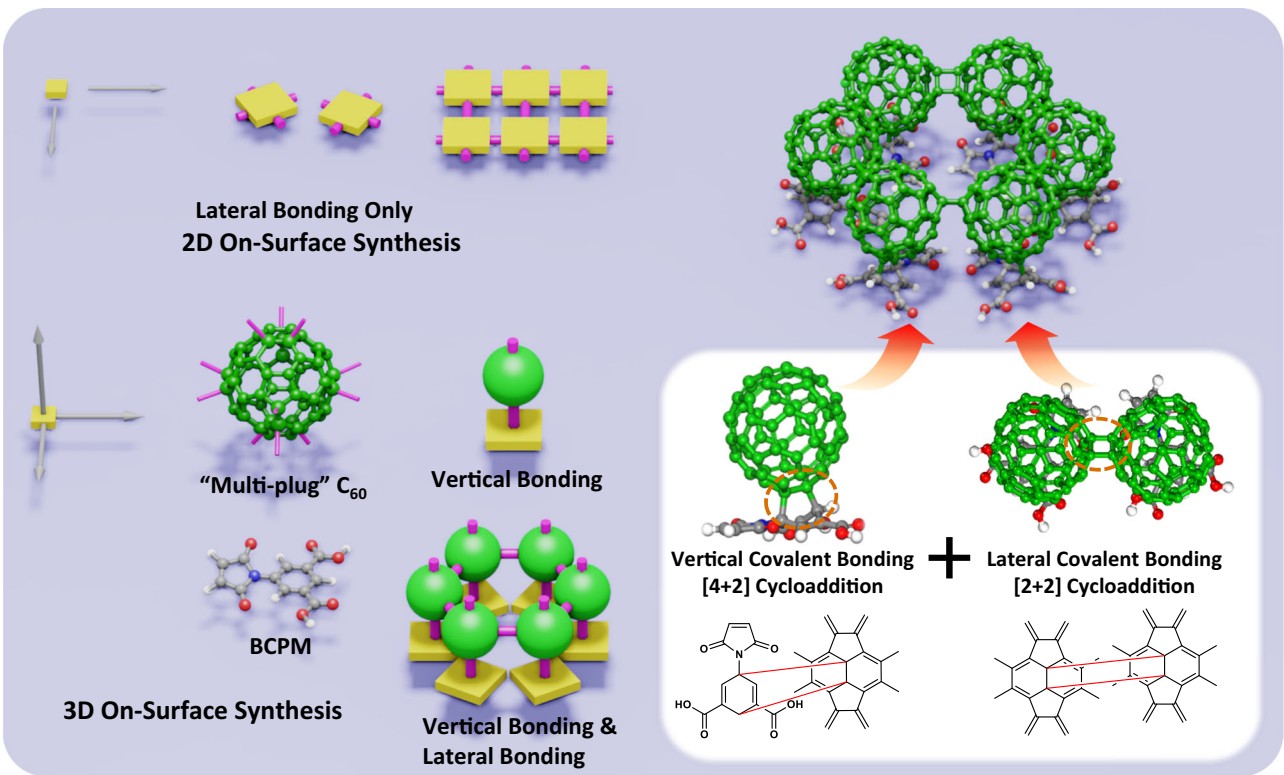

**Fig. 1 | Schematic illustration for the cycloaddition of $C_{60}$ and BCPM on Au(111).** Two-dimensional (2D) on-surface synthesis (OSS) is based on the dissociation/formation of organic precursors' σ bonds at the planar active sites of precursors, allowing only lateral covalent bonding within a single molecular layer thus confining the synthesis to be 2D. In addition to the lateral covalent bonding, to extend OSS from 2D to 3D, covalent bonding perpendicular to the surface is also required. In this work, 3,5-bis(carboxylic acid)-phenyl-3-maleimide (BCPM, $C_{12}H_7NO_6$) molecules assemble into a 2D ordered honeycomb network on Au(111), where carbon, nitrogen, oxygen, and hydrogen atoms are in grey, blue, red, and white,

respectively. The $C_{60}$ layer is constructed on the BCPM layer. One $C_{60}$ sits on top of the phenyl ring of one BCPM, providing appropriate steric configuration for the coupling between the lowest unoccupied molecular orbital of $C_{60}$ and the highest occupied molecular orbital of BCPM. Upon thermal activation, [4 + 2] cycloaddition between the phenyl ring of BCPM and [6,6] bond of $C_{60}$ is triggered. Thanks to the multiple reactive sites of $C_{60}$, the resultant $C_{60}$-BCPM molecules can laterally bond with one another by [2 + 2] cycloaddition between [6,6] bonds of their $C_{60}$ heads. In this way, both lateral and vertical covalent bonding is realized, representing a prototype for 3D synthesis on surfaces.

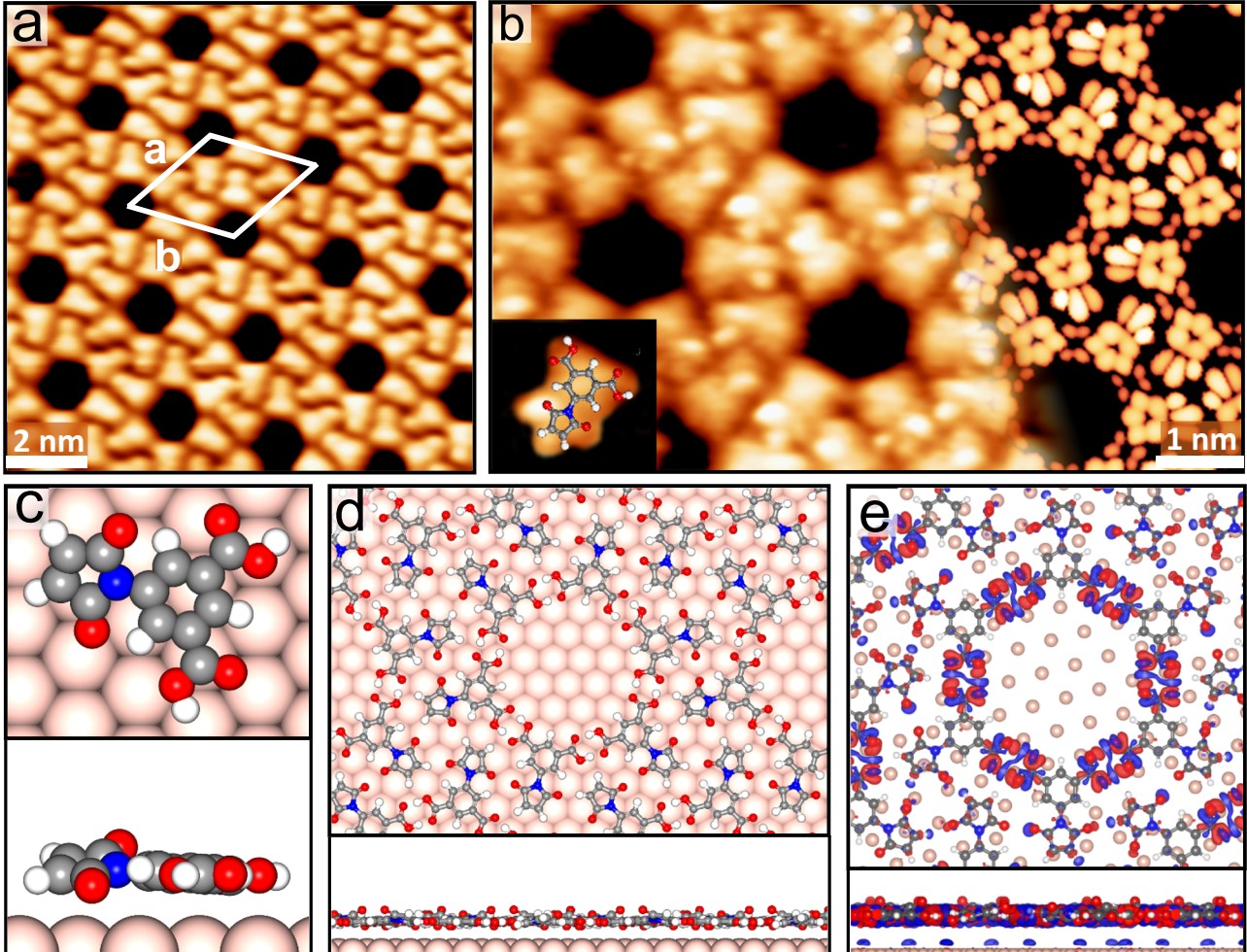

**Fig. 2 | Self-assembled BCPM network on Au(111). a** Scanning tunneling microscopy (STM) image of an extended network of BCPM on Au(111), where the unit mesh is marked by a white rhombus ($V_s$ = −1250 mV, $I_t$ = −0.27 nA). **b** Close-view STM image (left) of BCPM on Au(111) presenting the sub-molecular resolution, in good accordance with the density functional theory (DFT)-calculated image (right); the inset shows the morphology of a single BCPM superimposed by the BCPM structural model ($V_s$ = +1250 mV, $I_t$ = +0.66 nA). **c** Top and side views of DFT-optimized structural model of a single BCPM on Au(111), presenting the flatly-adsorbed bis(carboxylic acid)-phenyl group and the tilted maleimide group. **d** DFT-optimized structure and (**e**) electron density difference map of BCPM network on Au(111) (the isosurface value is 0.01 e/Å³), revealing that each BCPM molecule is anchored on the substrate with ligand bonding between the lower oxygen atom of its maleimide group and the substrate Au atom.

## Adsorption of $C_{60}$ on BCPM-coated Au(111)

$C_{60}$ molecules were then deposited onto the BCPM layer-covered Au(111) at room temperature (RT). At low dosages, $C_{60}$ adorbs preferentially in the circular pores of the BCPM network and is imaged as a bright circular protrusion (Fig. 3a, b). DFT-calculated structural model (Fig. 3c) reveals that the six pairs of double hydrogen-bonded bis(carboxylic acid)-phenyl groups in each BCPM unit create a ring-shape corral (diameter of 13.9 Å) to confine a spherical $C_{60}$ molecule in the center, with the center of $C_{60}$ at 6.1 Å above Au. Based on the electron density difference map (Fig. 3d), there is significant electron transfer from the Au substrate to the bottom atoms of $C_{60}$. Consistent with the STM images, $C_{60}$ molecules in the simulated image (Fig. 3e) show high contrast.

After all the pores of the BCPM network are occupied, further deposition of $C_{60}$ at RT (without post-annealing) leads to a coating of the $C_{60}$ layer on the BCPM layer. These $C_{60}$ molecules follow the same symmetry and periodicity as the underlying BCPM network (Supplementary Fig. 2) and form a configuration with one $C_{60}$ atop one BCPM ($C_{60}$-on-BCPM). The coverage increases with the $C_{60}$ dosage until the BCPM network is fully covered (Fig. 4a, b). The DFT-optimized model shows that each $C_{60}$ in the coating layer adsorbs exactly on top of the phenyl ring of one BCPM of the layer underneath, with $C_{60}$ center

10.1 Å above the Au(111) surface (Fig. 4c). The $C_{60}$ molecules on the BCPM layer are laterally confined by the tilted maleimide of BCPM and the guest $C_{60}$ in the pores of BCPM network. The calculated molecular arrangement for the extended network (Fig. 4d) and their molecular morphology (Fig. 4e) fit the STM results well.

## [4 + 2] cycloaddition between $C_{60}$ and BCPM layers

After annealing the $C_{60}$ layer on BCPM at 370 K for 30 min, two types of territories are observed (Fig. 5a). The brighter domain outlined by the cyan dashed line (D1) is distinct from the rest section (D2): i) although both show a similar close-packed arrangement as that of pristine $C_{60}$ on a bare Au(111)[28,29], linescans across the two domains show that $C_{60}$ in D1 is 3.5 Å higher than in D2 (Fig. 5b); ii) while D2 is rough with a large proportion of dim molecules and randomly-distributed bright ones, i.e., the typical structure for the disordered 2√3×2√3-R30° phase of $C_{60}$ on Au(111)[28], D1 is smooth, showing a uniform contrast; iii) surprisingly, even upon scanning at RT, distinct triangular sub-molecular feature of $C_{60}$ is observed from D1, suggesting that the $C_{60}$ are firmly frozen, adopting the same orientation (Fig. 5c). This is abnormal: intramolecular structure of pristine $C_{60}$ on Au(111) was only observed when quenching the molecules under cryogenic conditions;[29] at RT, pristine

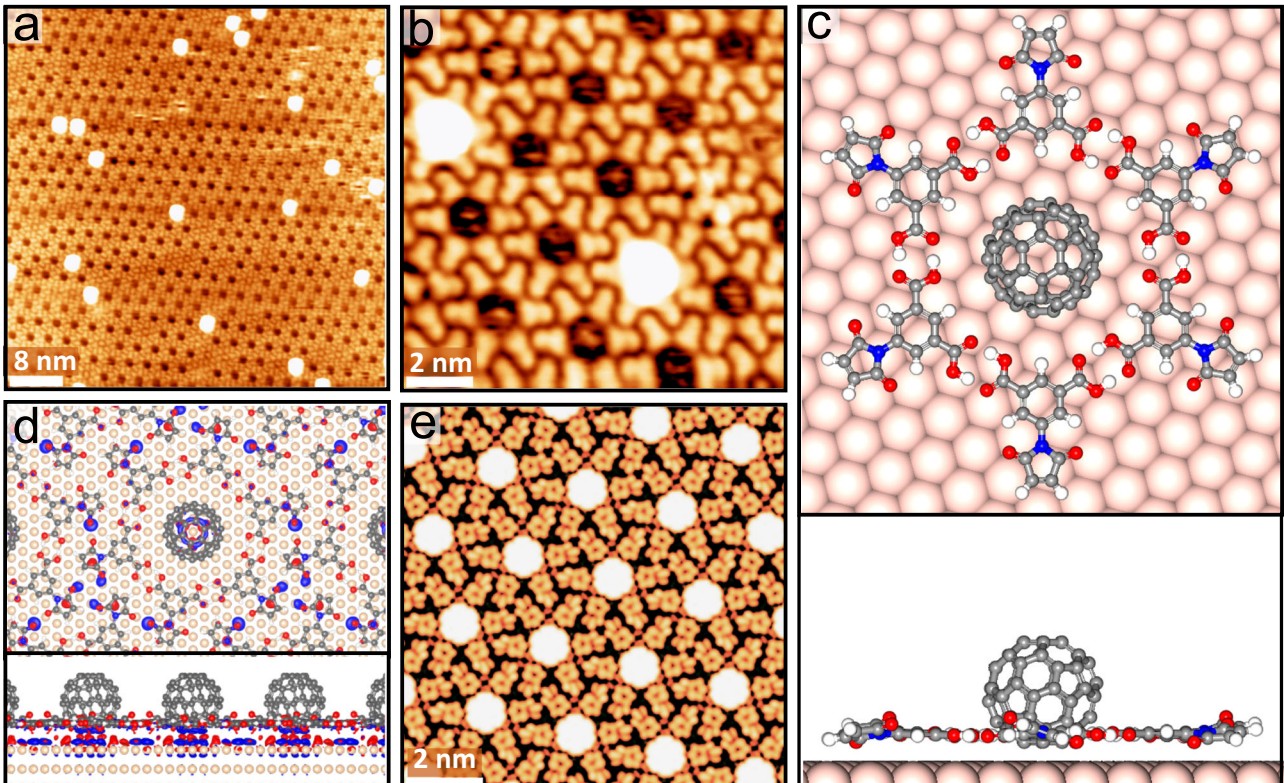

**Fig. 3 | Adsorption of $C_{60}$ on BCPM-coated Au(111) subjected to a small $C_{60}$ dosage. a** Large-scale ($V_s$ = +1250 mV, $I_t$ = +0.28 nA) and (**b**) Close-view ($V_s$ = −1250 mV, $I_t$ = −0.42 nA) STM images of the surface, showing a host-guest structure with $C_{60}$ located in the pores of the BCPM network. **c** Top and side-view structural model of a flower-like unit of six BCPM molecules hosting one $C_{60}$. **d** Electron density difference map (the isosurface value is 0.006 e/Å³), and (**e**), calculated STM image of the BCPM network with one $C_{60}$ in each pore.

$C_{60}$ molecules can only be imaged as featureless spheres on Au(111) due to their high rotational mobility at this temperature[40,41].

Given i) the greater height of molecules in D1 than that in D2, ii) the frozen morphology showing intramolecular features at RT, iii) the varied lateral arrangement of molecules compared with that before annealing (Fig.4), iv) and the presence of BCPM on the surface (the desorption temperature of BCPM from Au(111) of ∼ 420 K is much higher than the applied annealing temperature), and v) the favorable steric configuration of one $C_{60}$ on top of one BCPM, thermo-activated coupling between $C_{60}$ and BCPM is rationalized. Confirmed by the frontier molecular orbital analysis, the orbital matching and the appropriate steric arrangement can enable [4 + 2] cycloaddition between[6,6] bond at the bottom of $C_{60}$ and the C atoms in para-positions of BCPM phenyl ring, forming a covalently-bonded adduct, i.e., $C_{60}$-BCPM (Fig. 5d and Supplementary Fig. 3). Top of $C_{60}$-BCPM is calculated as being 3.4 Å higher than that of pristine $C_{60}$ on Au(111), consistent with the experiments. Upon [4 + 2] cycloaddition, hydrogen bonding between $C_{60}$-BCPM is largely weakened compared with that between pristine BCPMs (details in Supplementary Fig. 4) and can be easily dissociated upon annealing, allowing $C_{60}$-BCPMs to rearrange into D1 while the pristine $C_{60}$ (e.g. the guest $C_{60}$ in the pores of BCPM network) to self-assemble into D2. Meanwhile, the cycloaddition confines the rotation of $C_{60}$, allowing the sub-molecular resolution at RT. The molecular feature (Fig. 5e) and lateral arrangement of $C_{60}$-BCPMs in the calculated STM image (Fig. 5f) match the experimental results, where the observed triangle-shaped intramolecular feature is attributed to the top[6,6] bond of $C_{60}$-BCPM.

## Lateral covalent bonding of $C_{60}$-BCPMs

After forming $C_{60}$-BCPMs, the sample (as presented in Fig. 5) was further annealed at 490 K for 30 min. D1 remains the close-packed structure. Considering the low diffusion barrier of $C_{60}$-BCPM on Au(111) (Supplementary Fig. 5), the well-maintained lateral arrangement is attributed to the intermolecular interaction among $C_{60}$-BCPMs. Interestingly, bright macrocycles emerge in D1, where the hexamer rings (HRs) are dominant (Fig. 6a and Supplementary Fig. 6). Each HR is composed of six protrusions, showing $C_{6v}$ symmetry (Fig. 6b, c). Such macrocycles have been never observed from pristine $C_{60}$ on Au(111). Linescans show that the HR's diameter is 16.8 Å (Fig. 6d); the center-to-center distance between two neighboring protrusions in an HR is only 8.4 Å, which is largely reduced comparing with that of $C_{60}$-BCPMs in D1 before annealing at 490 K. The largely reduced spacing suggests that $C_{60}$-BCPMs, in this case, do not interact by vdW as would be the case of isolated molecules, but covalently bond with one another. Moreover, the height difference between the HR and the less bright protrusions (Fig. 6a, b) is much less significant than the height difference between the D1 and D2 domains. Both the small spacing of $C_{60}$-BCPMs and the small height difference exclude the possibility of forming the HRs by mixing D1 with D2. The robust bonding among $C_{60}$-BCPMs in the HR is also demonstrated by successive STM scanning with gradually increased current and decreased voltage at RT (Supplementary Fig. 7): the ring-like morphology composed of six molecules is well reserved when the tip is quite close to the surface. We then analyzed the bonding among $C_{60}$-BCPMs within the HR by DFT calculations. The possibility of single bond between $C_{60}$-BCPMs is ruled out: when a single bond is set in the initial structures of two $C_{60}$-BCPMs or the HR, the second bond forms after optimization, indicating that single bond between $C_{60}$-BCPMs is not stable (Supplementary Fig. 8). The optimized structure reveals that the geometry of $C_{60}$-BCPMs in the HR accepts [2 + 2] cycloaddition between[6,6] side bonds of $C_{60}$ part (Supplementary Fig. 8 and Fig. 6e), consistent with $C_{60}$ polymerization induced at high temperature and pressure[42,43]. The

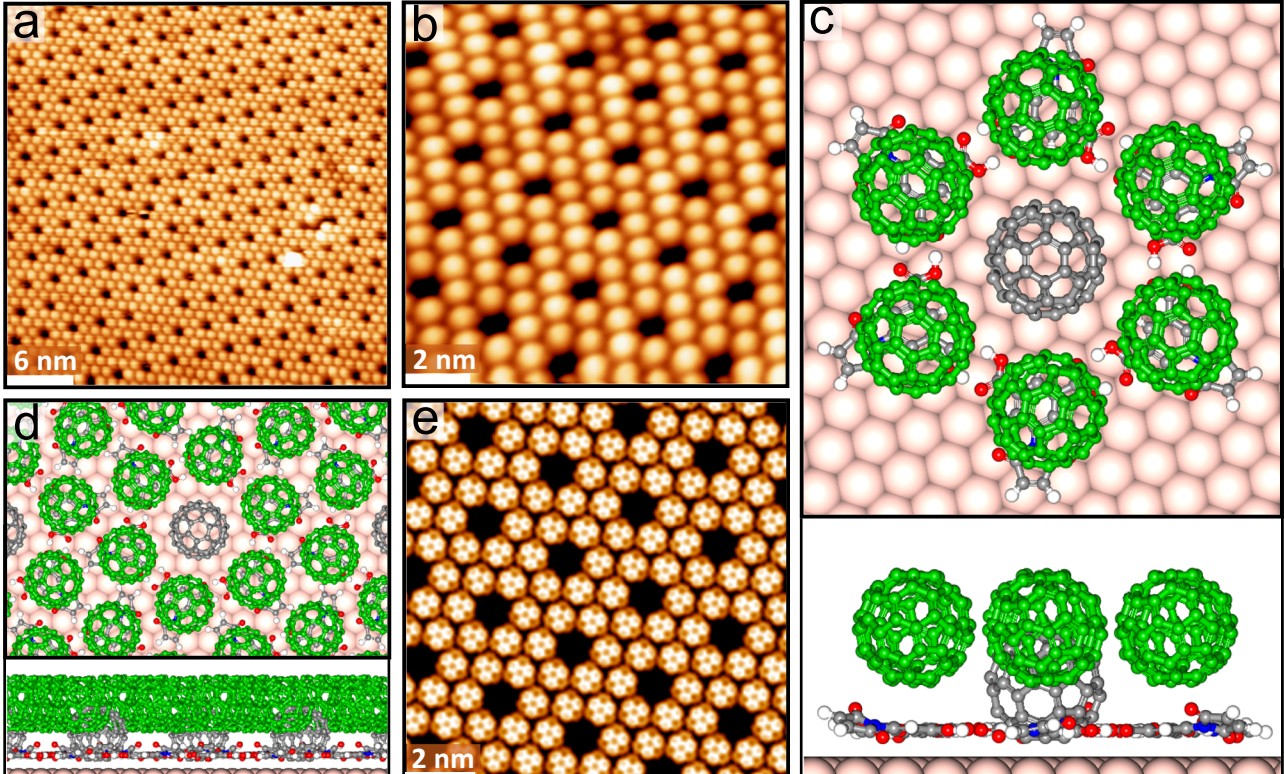

**Fig. 4 | Assembly of $C_{60}$ molecules on the BCPM layer on Au(111). a** Large-scale ($V_s$ = +1486 mV, $I_t$ = +0.75 nA) and (**b**) close-view ($V_s$ = +1250 mV, $I_t$ = +0.50 nA) STM images of the sample surface upon the further dose of $C_{60}$ after the BCPM pores are occupied, where the assembly of $C_{60}$ molecules presents the same symmetry and periodicity as the underlying BCPM network. **c** Top and side views of DFT-optimized structural model of the BCPM unit covered by $C_{60}$, where $C_{60}$ molecules sit on the phenyl rings of BCPM and are laterally confined by the tilted maleimide groups. **d** Top and side views of the DFT-optimized structural model, and (**e**) its calculated STM image of the extended $C_{60}$-on-BCPM network on Au(111). $C_{60}$ molecules on top of BCPM and in the network pore are shown in green and grey, respectively.

optimized structure, size, morphology, and contrast of HR composed of the six covalently-bonded $C_{60}$-BCPMs (Fig. 6f) are all consistent with the experimental results in Fig. 6a–c. This lateral [2 + 2] cycloaddition of $C_{60}$-BCPMs further distinguishes $C_{60}$-BCPM from pristine $C_{60}$. Moreover, the vertical [4 + 2] cycloaddition between $C_{60}$ and BCPM layers and this lateral [2 + 2] cycloaddition of $C_{60}$-BCPMs demonstrate the non-planar multiple covalent bonding capability of $C_{60}$ even upon OSS.

## Discussion

The interlayer [4 + 2] cycloaddition of $C_{60}$ and BCPM is distinct compared with the reported OSS strategy: the OSS reported previously (e.g. Ullmann, Glaser coupling) is based on the dissociation/formation of organic precursors' σ bonds at the terminal of molecular precursors, which can be achieved by intermolecular lateral coupling on surfaces[9–15], thus allowing 2D synthesis only; the present case forms covalent bonding between adjacent molecular layers based on the rehybridization of π orbitals of the reactants. Such cycloaddition is also different from that of BCPM with graphene, where cycloaddition occurs between C = C of maleimide of BCPM and graphene, involving both [2 + 2] and [4 + 2] cycloaddition with different pairs of carbon atoms of graphene hexagons[38]. In the present system, cycloaddition between $C_{60}$ and BCPM maleimide is not observed. Although the energy barrier of their [2 + 2] cycloaddition in the gas phase (1.3 eV) is only 0.1 eV higher than that of [4 + 2] cycloaddition between $C_{60}$ and BCPM phenyl ring (Supplementary Fig. 3 and Supplementary Fig. 9), the tilted maleimide on Au(111) makes its cycloaddition with $C_{60}$ unfavorable.

We performed climbing image nudged elastic band (CI-NEB) calculations to understand the cycloaddition pathways. For [4 + 2] cycloaddition between BCPM and $C_{60}$ on Au(111) (Supplementary Fig. 10a), the two bonds between the $C_{60}$[6,6] bond and the two carbon atoms at the para-positions of BCPM phenyl ring (marked as A and B, respectively) are not formed synchronously: C–C bond with Carbon A on the BCPM's phenyl ring forms first (transition state, TS), and then the second C–C bond with Carbon B forms (final state, FT); this is different from cycloaddition of $C_{60}$ with benzene where the two bonds of [4 + 2] cycloaddition form simultaneously. In TS, Carbon A and B locate at 3.6 Å and 4.3 Å above the Au(111), respectively, i.e., being lifted by 0.2 Å and 0.9 Å compared with the geometry of pristine BCPM on Au(111). The phenyl ring after cycloaddition becomes non-flat and distorted. We compared the energy barrier for cycloaddition of $C_{60}$ with benzene, benzene-1,3-dicarboxylic acid, N-phenylmaleimide, and BCPM (Supplementary Fig. 11), revealing that the equipped carboxylic acid and maleimide groups on the phenyl ring of BCPM can decrease the [4 + 2] cycloaddition barrier. Moreover, the Au(111) surface further reduces the reaction barrier from 1.28 eV in the gas phase to 1.21 eV.

For the lateral [2 + 2] cycloaddition between the[6,6] side bonds of $C_{60}$ of two $C_{60}$-BCPMs (Supplementary Fig. 10b), the barrier is 1.87 eV (0.66 eV higher than that of the [4 + 2] cycloaddition). According to DFT calculations, the HOMO-LUMO gap of $C_{60}$-BCPM is 0.23 eV smaller than that of $C_{60}$ and the energy barrier of [2 + 2] cycloaddition (in gas phase) for $C_{60}$-BCPM is also 0.16 eV lower than that of $C_{60}$ (Supplementary Fig. 12). In this regard, [2 + 2] cycloaddition among $C_{60}$ parts of $C_{60}$-BCPMs is promoted compared with the case of pristine $C_{60}$.

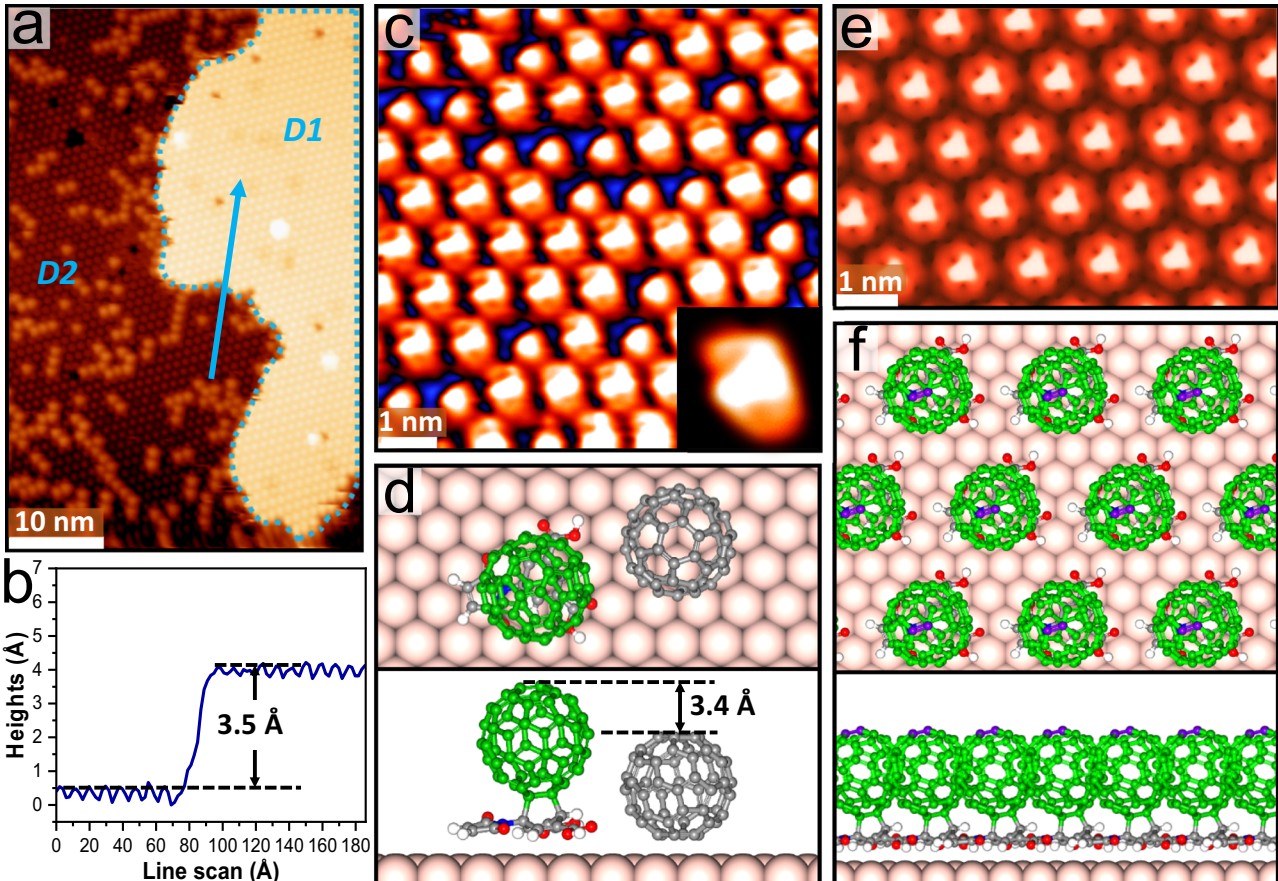

**Fig. 5 | Interlayer [4 + 2] cycloaddition between $C_{60}$ and BCPM on Au(111).**
**a** Large-scale STM image after annealing the $C_{60}$-on-BCPM network at 370 K for
30 min, showing two types of territories where D1 is the regular and bright domain
outlined by the cyan dashed line and D2 represents the dark and less ordered
section ($V_s = -1250$ mV, $I_t = -0.28$ nA). **b** Linescan across the two territories marked
by the blue arrow in panel a, showing a height difference of 3.5 Å ($V_s = -1250$ mV,

$I_t = -0.28$ nA). **c** High-resolution STM image of D1, showing the intramolecular
structure of $C_{60}$. **d** Top and side views of the structural model of $C_{60}$-BCPM and
pristine $C_{60}$ (grey) on Au(111), showing the height difference consistent with the
linescan. **e** The calculated STM image and (**f**) structural model of the close-packed
$C_{60}$-BCPM structure, i.e., D1, on Au(111), where the top [6,6] bonds highlighted in
purple correspond to the observed triangle-shaped intramolecular feature.

As demonstrated by our experimental results and DFT calculations, thanks to the multiple reactive sites of $C_{60}$, both the vertical covalent bonding via interlayer [4 + 2] cycloaddition between $C_{60}$ and the phenyl ring of BCPM and the lateral covalent bonding between the $C_{60}$ heads of $C_{60}$-BCPMs via [2 + 2] cycloaddition are constructed, representing a prototype of 3D synthesis on the surface. We further considered the possibility of constructing BCPM-$C_{60}$-BCPM and $C_{60}$-BCPM-$C_{60}$ covalently-bonded sandwich for BCPM-$C_{60}$-BCPM-$C_{60}$ repeated multiple layers. Based on the DFT calculations (Supplementary Fig. 13), $C_{60}$ with its bottom already bonded with BCPM is still able to react with BCPM atop of it by cycloaddition; BCPM with its phenyl/maleimide ring bonded with $C_{60}$ can still offer the other free conjugated ring to have cycloaddition with $C_{60}$. Following this pattern, 3D covalently-bonded organic architectures/devices with extended/controlled thickness could be bottom-up fabricated via cycloaddition reactions in principle, employing fullerenes and organic compounds with multiple conjugated rings like Lego pieces with two connectors. Considering the less surface-confinement effect hence higher freedom of the molecules as well as the possible unfavorable orientations of the formed isomers, the experimental realization may be challenging, requiring fine control and further exploration.

3D covalent bonding has been realized by cycloaddition of $C_{60}$ on Au(111). The carboxylic acid groups of BCPM enable the formation of

a well-defined template for $C_{60}$, while the out-of-plane maleimide group enables strong adsorption of the molecules on the substrate. The configuration of $C_{60}$ on BCPM allows appropriate molecular orbital hybridization between them meanwhile the tilted maleimide group of BCPM restrains the lateral diffusion of $C_{60}$, enabling covalent bonding perpendicular to the surface via interlayer [4 + 2] cycloaddition upon thermal activation. The cycloaddition is demonstrated by the increased adsorption height, the distinct domain structure, and the frozen orientation hence the triangular submolecular feature at RT, distinct from pristine $C_{60}$. Moreover, the rotation-frozen $C_{60}$ in $C_{60}$-BCPMs, their close-packed molecular arrangement, and the multiple non-planar reactive sites of $C_{60}$ allow, upon elevated annealing, a next-step lateral [2 + 2] cycloaddition of $C_{60}$-BCPMs to form macrocycles, further distinguishing the difference of $C_{60}$-BCPM and pristine $C_{60}$. Moreover, both $C_{60}$ and BCPM are able to have a second cycloaddition along the direction perpendicular to the surface. Allowing both vertical and lateral covalent bonding, this work unlocks an efficient route for bottom-up synthesis of covalently-bonded 3D organic framework on surfaces via cycloaddition of organic compounds with fullerenes--the steric multiplug reactants, thus extending OSS from 2D to 3D. The strategy of interlayer covalent bonding may innovate the design of robust heterolayered materials for engineering cutting-edge devices with molecular precision.

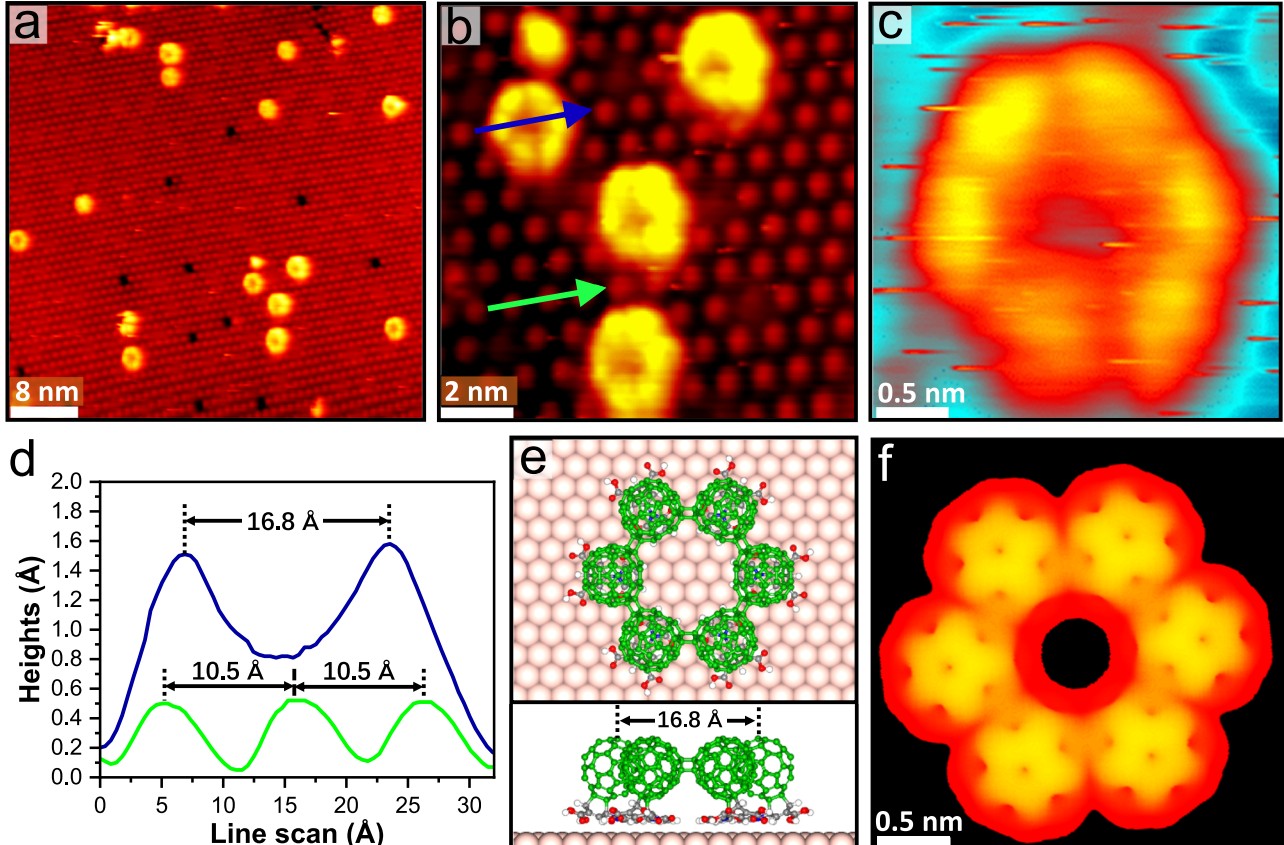

**Fig. 6 | Lateral [2 + 2] cycloaddition of $C_{60}$-BCPM. a** Large-scale ($V_s$ = −1250 mV, $I_t$ = −0.26 nA) and (**b**), close-view ($V_s$ = −1250 mV, $I_t$ = −0.29 nA) STM images showing the hexamer rings (HRs) after annealing $C_{60}$-BCPM on Au(111) at 490 K for 30 min. **c** High-resolution STM of an individual HR. **d** Linescans along the blue and green arrows in (**b**), showing the size of the HR compared with the periodicity of $C_{60}$-BCPM in D1. **e** Top and side views of the DFT-optimized structure of an HR on Au(111), where the six $C_{60}$-BCPM molecules are covalently bonded via [2 + 2] cycloaddition. **f** Calculated STM image of an HR, showing consistent morphology as observed experimentally.

## Methods

### Experimental details

All experiments were performed in an ultrahigh vacuum (UHV) chamber equipped with a variable-temperature Aarhus scanning tunneling microscope. The typical base pressure of the UHV chamber was $1 \times 10^{-10}$ mbar. The Au(111) substrate was cleaned by repeated cycles of argon ion sputtering (1000 eV) and annealing (at 770 K). BCPM compound and $C_{60}$ were sublimated from the crucibles of a low-temperature evaporator at 330 K and 470 K, respectively, onto the surface kept at RT. All STM images were collected in the constant current mode at RT.

### Calculation methods

DFT calculations were performed using the Vienna ab initio simulation package (VASP)[44]. The interaction between electrons and ion cores was described by the projected augmented wave (PAW) method[45] with a plane-wave cutoff energy of 400 eV. The exchange and correlation effects between electrons were treated by Perdew-Burke-Ernzerhof (PBE) exchange-correlation density functional[46]. The vdW correction was performed using the zero damping DFT-D3 method of Grimme[47]. The BCPM network, BCPM network with $C_{60}$ hosted in the pores of that network, and the steric assembly of $C_{60}$ on atop of BCPM were calculated in a 9 × 9 supercell on the Au(111) substrate. The close-packed arrangement of the $C_{60}$-BCPM adduct was calculated in a $2\sqrt{3} \times 2\sqrt{3}$ – R30° supercell on Au(111). The hexamer ring composed of six $C_{60}$-BCPMs was calculated in a 16 × 16 supercell on Au(111). The CI-NEB calculations[48] for [4 + 2] cycloaddition between $C_{60}$ and BCPM and [2 + 2] cycloaddition between the $C_{60}$-BCPMs were performed in a 9 × 9 supercell on Au(111), with 12 and 11 images inserted between the initial and final states, respectively. The Brillouin zone was sampled through the Monkhorst-Pack scheme with 3 × 3 × 1 mesh for the $2\sqrt{3} \times 2\sqrt{3}$-R30° supercell and only Gamma point for the other cells with lattice parameters larger than 20 Å. In the structural relaxation, all structures were optimized until the residual forces were found smaller than 0.02 eV/Å. In the CI-NEB calculations, the force criterion was 0.05 eV/Å for each image. The Tersoff-Hamman method was employed for the calculations of the STM images[49].

## Data availability

The authors declare that all data supporting the findings of this study are available within the paper [and its supplementary information files]. All other data supporting the study are available from the corresponding author upon request. Correspondence and requests for materials should be addressed to M.Y.

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

## Acknowledgements

This work is financially supported by the National Natural Science Foundation of China (21473045, M.Y.; 51772066, Y.S.; 52073074, Y.S.;

22272041, M.Y.), and State Key Laboratory of Urban Water Resource and Environment, Harbin Institute of Technology (2021TS08). F.R. is grateful to the Canada Research Chairs program for partial salary support.

## Author contributions

S.W., P.D., and M.Y. designed the project; C.M. and A.G. synthesized the BCPM compound; P.D., S.W., Z.L. and G.S. conducted the assembly/reaction of C60 and BCPM on Au(111)/STM imaging; P.D. and L.K. performed the calculations; P.D. and M.Y. analyzed and interpreted the results; P.D., M.Y., Y.S., L.K., F.R. and F.B. wrote/revised the manuscript.

## Competing interests

The authors declare no competing interests.

## Ethical approval

This manuscript complies with the Ethics and inclusion guidelines of the Nature Portfolio Journals.
