## [Peer Review File · Nature Communications]

Extending on-surface synthesis from 2D to 3D by cycloaddition with C60Reviewers' Comments:

Reviewer #1:

Remarks to the Author:

This paper describes the surface-confined formation of 2D polymers by using on-surface synthesis. The authors investigated the formation of two supramolecular networks on an Au(111) surface and their transformation upon thermal annealing by using STM. The experimental part of this manuscript is well-written and detailed. Concerning the novelty, this manuscript is different from the paper published in 2020 (Nat Chem, ref 36 in the present manuscript) by some of the authors. However, I have a major concern as regards the authors claiming that their achieving on-surface three-dimensional synthesis.

The title of the article is "Extending on-surface synthesis from 2D to 3D by cycloaddition with C60." Still, the present manuscript submitted by Yu et al. actually describes the formation of a 2D supramolecular network, then the deposition of a 3D molecule (C60) and finally the formation of covalent bonds between the underlying 2D supramolecular network and the deposited C60 molecules. The 3D growth is only discussed as one possible perspective (line 224), supported by two DFT calculations which are described in the supporting information (Fig. S10).

The reference 1 of the manuscript of Yu et al., which is an article published in 2011 by Blunt et al. in Nature Chemistry, describes: "Two-dimensional porous arrays provide host sites for trapping guest species of suitable size. Here we show that a non-planar guest species (C60) can play a more complex role by promoting the growth of a second layer of host molecules (p-terphenyl-3,5,3',5'-tetracarboxylic acid) above and parallel to the surface so that self-assembly is extended into the third dimension." In the article of Blunt et al., there is an experimental demonstration that one can create a second layer by using the first underlying system, at the supramolecular level (i. e. without any covalent bond).

In the article of Yu et al, the demonstration of the extension of 2D to 3D by the formation of covalent bonds is only suggested in supporting information (Fig. S10) and this claim is not supported by the described experimental results. Moreover, the suggested mode of 3D growth (Fig. S10) is not well described because once the BCPM molecule will have reacted with the C60 by cycloaddition, it is not explained how another C60 molecule will be able to react again with BCPM molecule to promote the 3D growth.

Therefore, the only 3D character described in this article is justified by the 3D nature of the C60 used. This result is not in accordance with the title, the abstract, the introduction and the conclusion of the article where the authors claim that their major result is "extending OSS from 2D to 3D" and that "this work unlocks an unconventional route for...3D covalently-bounded...surfaces" (lines 30-31). That claim is definitely not supported by the experimental results described in this manuscript, while it is a key point for the acceptance of an article.

Outside this major point, a few minor comments can also be made to the manuscript:

- 1) The imaging conditions (temperature, current, voltage) are missing from the captions of the figures describing the STM images. These experimental data are very important to be able to compare them with each other.
- 2) C60 molecules adsorbed in the pores of the supramolecular network constituted by BCPM molecules are bright (Fig 3a-b) before the cycloadditions while they become dark (Fig 4a-b) after this reaction. Could the authors explain this difference?
- 3) Have the authors investigated the effect of thermal annealing at 370 K for 30 min on the BCPM supramolecular network without C60 molecules?
- 4) If I understand well the discussion on the reactivity in the investigated cycloaddition (lines 207-221), the BCPM molecules act as the dienes, reacting via their HOMO, while the C60 molecule plays the role of a dienophile, reacting via its LUMO (Fig S3). It seems, from Figure S9, that the addition of electron-withdrawing groups (COOH) on the BCPM molecule (diene) promotes the reaction. However, it is well known that usual [4+2] cycloadditions, which are concerted reactions, are promoted when

electron-donating substituents are added to the diene and electron-withdrawing groups to the dienophile. Can the authors explain this surprising effect in more details?

Reviewer #2:

Remarks to the Author:

In this manuscript, Miao Yu et. al developed an interlayer covalent reaction by [2+4] cycloaddition between [6,6] bond of C60. The morphology change upon sublimation of BCPM, C60, and interlayer [2+4] cycloaddition were thoroughly characterized and has a reasonable match with DFT calculation. The resulting 3D covalently-bonded organic architectures with C6v symmetry is also of interests to chemistry and organic electronics community. Overall, the synthesis method is simple while efficient, the characterization is comprehensive. I suggest to be published in nature communication after these comments have be addressed:

1. Fullerene derivatives are known to have poor thermostability so that fewer materials have been used for vacuum-deposition process. Please show the TGA result of pristine C60, BCPM, and resulting C60-BCPM to show the thermostability at 490 K, at which temperature authors checked the reactivity of C60-BCPM.
2. To prove the novelty of the 3D structure formed by interlayer covalent, please also attach the test result of direct sublimation of C60-BCPM onto Au(111) surface.

Reviewer #3:

Remarks to the Author:

Please check the attached pdf file.

General comments

The present manuscript reports the on-surface synthesis (OSS) of 3D molecular architectures (i.e., structures with more than one molecular layer in height) by exploiting the cycloaddition reaction between C60 and BCPM at the surface of Au(111). The paper is concise and well-structured. The results are generally well discussed, and the authors try to explain the experimental observations. The development of surface built-in 3D molecular architectures represents a significant advancement in surface science. I consider this work an important contribution to the field, although I believe it can still be improved in its scientific content and presentation. Although the experimental results constitute good proof of what is happening at the surface, I am skeptical about some interpretations of the phenomena.

Pros: Clear explanation of the various steps of the work. The steps of the on-surface synthesis (OSS) are generally well-described and adequately supported by the experimental data. The main idea of the paper is very promising and relevant for the field. The observation of the fullerene hexamers in the surface is particularly interesting!

Cons: The interpretation of the results is somewhat lacking in what concerns bonding among the intervening species. Some apparent misconceptions about cycloaddition reactions.

After careful consideration and reviewing of the manuscript I recommend publication in *Nature Communications* after some major revisions noted.

Major comments (publication is not recommended before addressing the following issues)

1. The bonding between C60 and BCPM on the Au(111) surface

Some details of the theoretical study of the [4+2] reaction need some clarification. Firstly, the authors admit that the reaction involves the HOMO of the diene and the LUMO of the dienophile. According to Supp. Fig. 3, however, the situation is not that clear, because there is a smaller energy gap between the LUMO of the diene and the HOMO of the dienophile. This is consistent with the presence of the π -electron withdrawing -COOH groups in the diene. In these cases, an inverse Diels-Alder cycloaddition can occur, involving the LUMO of the diene and the HOMO of the dienophile. The authors should clarify this point. My second concern are the calculated energies for the [4+2] reactions. Cycloaddition reactions are known to be substantially exothermic; however, the

results depicted in Supp. Figs. 8a and 9 suggest otherwise (the calculated reactions are highly endothermic, around +90 kJ/mol). In the light of these results, one may ask what is the driving force for this reaction (it is enthalpically and entropically disfavored...)? Is there any problem with the calculations? The DFT-D3 method seems adequate for these systems with considerable dispersion forces. Have the authors considered the correction for the basis-set superposition error (BSSE)? For example, the initial states in Supp. Figs. 8a and 9 can be erroneously stabilized by BSSE due to the formation of the van der Waals complex between the diene and dienophile. It is true that in all systems considered in Supp. Fig. 9 (including BCPM) there is a significant breaking of aromaticity in the diene upon cycloaddition, which contributes for making the reaction less exothermic. With all these concerns, I'm not sure if the reaction occurring at the surface between C60 and BCPM is the [4+2] cycloaddition proposed by the authors. I'm not an expert of STM, but I think this technique cannot provide many details about the nature of the bonding occurring at the surface. The location and structure of the molecules above the surface are clear, however, the way the molecules bond among each other is open to speculation. Given this, I ask if there are other bonding models compatible with the experimental observations? The authors should review the calculations to check for eventual errors in the calculated energies. Additionally, they should provide more experimental evidence for the type of reaction occurring between C60 and BCPM. For instance, they could make a reaction control test between C60 and BCPM in solution and characterize the product. Has this been done by other authors before?

2. The (im)possibility of a thermal [2+2] cycloaddition

Although the experimental observations give strong support for the formation of macrocycle hexamer rings after annealing at 490 K, I have some doubts about the provided rationalization. The authors state that the various C60 in the hexamer are covalently bonded via [2+2] cycloaddition. However, [2+2] cycloadditions are generally thermally forbidden and need to be photochemically induced to occur (with rare exceptions). In the conditions reported by the authors I am not convinced that the C60 molecules in the hexamer react via [2+2] cycloaddition. Actually, solid C60 is stable at 490 K, and no covalent bonding between the various molecules in the crystal is observed. Upon heating to 490 K the molecules in the crystal remain covalently separated, like in a regular crystalline solid. The STM images are consistent with the presence of the hexamer but furnish little information about the bonding nature among the C60s. There are other possibilities for explaining this bonding, as for example single, or sp²-like bond between the C60. Alternatively, and perhaps more probable, the hexamers are just the parent structures of C60 sitting above the original BCPM flower structure. Heating at 490 K shall provide enough thermal energy for the C60

molecules to acquire significant surface mobility and become more scarce and less structured. The resulting hexamers are possibly just the remaining structures that survived heating. In page 9, line 175 the authors say that “the sample was annealed at 490 K for 30 min” but do not give important details about this sample. I presume that it is a sample like the one depicted in Fig. 5 a (after annealing the C60-BCPM network at 370 K). In that case, the observed hexamers after heating at 490 K can be the result of a D1 territory dispersing through a D2-like one (or similar). In page 9, lines 178-182 the authors do refer that the hexamers have a smaller radius than the analogous structures observed in D1 motifs. However, without experimental uncertainties one cannot judge if this difference is statistically significant. Also, the authors are comparing the diameter of the C60 hexamers with the layer below (Fig. 6 b and d), which I’m not sure what it is... Is it Au, the network of BCPM or the similar C60-BCPM layer observed in D1? Note that the position of the hexamers above the background layer does not seem very plausible with the presence of a D1 structure below or side-by-side. Comparisons should be made between the C60 hexamer rings (Fig. 6 b) with those similar structures observed in D1 motifs (Fig. 5 c). Although the authors say it correctly in the Fig. 6 caption (for panel d), the images are confusing and I’m not sure if the green arrow corresponds to a D1 motif. The authors should clarify this point and describe in more detail the resulting surface after annealing at 490 K. Even if these diameters are proven to be different, that does not prove conclusively the existence of covalent bonding among the C60s in the hexamers. The smaller diameter in the hexamers (Fig. 6 b) can be due to other factors, like the absence of neighboring fullerenes or of the central physisorbed C60 (the grey C60 in Fig. 4). If a [2+2] cycloaddition occurs the authors should provide more convincing experimental proofs. The authors should also provide more details about the sample subjected to annealing at 490 K and prove (or disprove) the hypothesis of thermal diffusion through the surface.

3. Uncertainties associated to STM measurements

The authors should indicate the calculated uncertainties (if not possible, an educated estimation), associated to the distance measurements by STM.

Minor comments (following these comments shall improve the quality of the manuscript in its presentation and exactness)

1. The use of the notation “[2+4] cycloaddition”

All through the manuscript, the authors refer to the [4+2] cycloaddition reactions as [2+4]. Although this is not incorrect, the accepted notation is [4+2] (4 π electrons from the diene and 2 π electrons from the dienophile). Please correct the notation.

2. Page 3, lines 57-60, the concept of a 3D molecule

The authors speak of the concept of a 3D molecule. What is exactly meant by a “3D molecule”? Why is C60 the most representative “3D molecule”? Every molecule can be considered 3D... The authors also refer “unique 3D conjugated electronic structure”. The use of the term “3D” in these contexts is confusing. Please explain, or just remove this.

3. Page 3, line 57, properties of C60

Saying that “C60 has magnificent physical properties/potentials” is misleading, because, for example, C60 as a single molecule does not present superconductivity. Some properties are specific to a particular state of matter, not necessarily inherent to the molecule. Moreover, many times such magnificent properties are reported for functionalized C60, not C60 itself. This sentence, as it is, is a bit vague. Please be more specific of which state/form of C60 display the properties referred.

4. Page 3, lines 65-66, and in more instances of the manuscript – the idea of spatial requirements for the reaction with C60

In some parts of the manuscript (page 3 line 65; page 3 line 69; page 4 line 84; page 8 line 163; page 12 line 239), the authors imply that C60 and BCPM have an appropriate steric configuration for the reaction. This is not surprising given the spherical symmetry of C60 and the fact that BCPM is immobilized at the surface. Hence, this reaction shall not be particularly characterized by orientation requirements of the reacting molecules, other than the well-known spatial requirements of the overlapping orbitals in the diene and dienophile in [4+2] cycloadditions. In my opinion, the idea that should be conveyed here is that adsorbed BCPM has a geometry that allows it to act like a diene in a subsequent reaction with C60. On the other side, due to the high symmetry of C60 (every [6,6] bond is equally reactive), there are actually no significant orientational requirements for it to react with BCPM. I understand what the authors mean, but, as it is, it may pass the idea that C60 must approach BCPM with a very specific orientation. Given the nature of the system under study

(immobilized BCPM and symmetrical C60) it is easily recognized that spatial requirements will be a minor issue for the reaction; there is no need to emphasize it so much in the manuscript.

5. Page 11, lines 221-223

Where the authors say “We believe that this is related to the rotation of pristine C60 molecules at RT (or a higher temperature) [38,39], making the covalent coupling between neighbouring C60 impossible.” Rotation of a molecule does not prohibit the formation of covalent bonds. If this was the case, reactions in solution, liquid and gas phase would be impossible. Please correct it.

6. Page 11, lines 224-233, the possibility of growing the 3D motif

The discussion of further evolving the 3D motif of fullerene-BCPM layers above the Au surface is very interesting and relevant for the discussion. The authors correctly recognize that this is probably experimentally challenging and refer the less surface-confinement effect and higher freedom of molecules. However, I see one important difficulty for this task that should also be mentioned here: the possibility that reactions on C60 after the first coupling with BCPM (or any other addend) can produce many possible isomers, many of which will not have an adequate orientation to grow the 3D structure. The authors only consider the more obvious trans-1 isomer (Supp Fig. 10 b), but many more are possible. Achieving high isomer selectivity in these syntheses is not an easy task.

7. Small typos and imprecisions along the manuscript

- Page 3, line 62, change “denes” to “dienes”.
- Page 3, line 75, change “forming macrocycles with the hexamers rings with C6v symmetry dominated” to “forming macrocycles mostly consisting of hexamer rings with C6v symmetry”.
- Page 7, line 142, change “two types of territories is observed” to “two types of territories are observed”.
- Page 8, line 156, where the authors say “showing the intramolecular structure of C60” did they mean intermolecular? Intramolecular implies covalent bonding among the molecules. The term “intermolecular is more correct in this context, since it describes the organization of various C60 molecules interacting via intermolecular interactions, like in the crystal phase.

Reviewer: #1

Comments:

This paper describes the surface-confined formation of 2D polymers by using on-surface synthesis. The authors investigated the formation of two supramolecular networks on an Au(111) surface and their transformation upon thermal annealing by using STM. The experimental part of this manuscript is well-written and detailed. Concerning the novelty, this manuscript is different from the paper published in 2020 (Nat Chem, ref 36 in the present manuscript) by some of the authors. However, I have a major concern as regards the authors claiming that their achieving on-surface three-dimensional synthesis.

Answer: We are grateful to the reviewer for taking the time to evaluate our manuscript and providing positive remarks and constructive comments. Following this reviewer's suggestions, we have included additional explanations and a more detailed analysis in the revised manuscript. Please refer to our point-by-point responses below.

1. The title of the article is "Extending on-surface synthesis from 2D to 3D by cycloaddition with C₆₀." Still, the present manuscript submitted by Yu et al. actually describes the formation of a 2D supramolecular network, then the deposition of a 3D molecule (C₆₀) and finally the formation of covalent bonds between the underlying 2D supramolecular network and the deposited C₆₀ molecules. The 3D growth is only discussed as one possible perspective (line 224), supported by two DFT calculations which are described in the supporting information (Fig. S10).

Answer: We are grateful to the reviewer for raising this crucial point. As described in the introduction, all the products obtained by on-surface synthesis (OSS) were based on lateral covalent coupling, which limits OSS to 2D and the products to a single-layer molecular thickness. Although multilayer supramolecular architectures have been proposed, these assemblies based on noncovalent bonding often have lower thermal stability, conductivity and mechanical properties compared with covalently-bonded products. In this work, we provide a breakthrough for OSS by pursuing interlayer covalent coupling, yielding the multilayer covalent bonding of 3D construction meanwhile keeping the high degree of order and high precision of OSS.

The point of 3D synthesis is demonstrated experimentally based on both [4+2] cycloaddition between C₆₀ and BCPM and [2+2] cycloadditions of C₆₀-BCPMs. In this way, both vertical and lateral covalent coupling are enabled in this system, presenting a prototype of 3D synthesis. In addition, we have investigated the double cycloaddition between C₆₀ and BCPM by DFT calculations and described/discussed the results in both main text (Page 12) and Supporting Information (Supplementary Fig.13). The results confirm that BCPM and C₆₀ can form multiple layers of covalently-bonded BCPM-C₆₀-BCPM-C₆₀, where the phenyl ring and maleimide ring of BCPM can provide double sites for

cycloaddition with C_{60} , and the multiple [6,6] bonds of C_{60} allows it to react with BCPM at both the top and bottom [6,6] bonds by cycloaddition. Following this pattern, 3D covalently-bonded organic architectures/devices with extended/controlled thickness could be fabricated bottom-up *via* cycloaddition reaction like Lego pieces with two connectors.

Following this reviewer's suggestion, we have addressed the point of 3D synthesis more clearly in the revised manuscript (Page 12) as follows:

“From the experimental results and DFT calculations, the interlayer [4+2] cycloaddition between C_{60} and BCPM and the intermolecular [2+2] cycloaddition among C_{60} -BCPMs demonstrate that both vertical and lateral covalent bonding involved in the present system, representing a prototype of 3D synthesis. We then considered the possibility of constructing BCPM- C_{60} -BCPM and C_{60} -BCPM- C_{60} covalently-bonded sandwich for BCPM- C_{60} -BCPM- C_{60} repeated multiple layers. Based on the DFT calculations (Supplementary Fig. 13), C_{60} with its bottom already bonded with BCPM is still able to react with BCPM atop of it by cycloaddition; BCPM with its phenyl/maleimide ring bonded with C_{60} can still offer the other free conjugated ring to have cycloaddition with C_{60} . Following this pattern, 3D covalently-bonded organic architectures/devices with extended/controlled thickness could be bottom-up fabricated via cycloaddition reactions in principle, employing fullerenes and organic compounds with multiple conjugated rings like Lego pieces with two connectors.”

“Allowing both vertical and lateral covalent bonding with extended dimension, this work unlocks an efficient route for bottom-up synthesis of covalently-bonded 3D organic framework on surfaces by employing cycloaddition of fullerenes and molecules with polyaromatic rings, and provides a practical solution for extending OSS from 2D to 3D.”

*2. The reference 1 of the manuscript of Yu et al., which is an article published in 2011 by Blunt et al. in Nature Chemistry, describes: “Two-dimensional porous arrays provide host sites for trapping guest species of suitable size. Here we show that a non-planar guest species (C_{60}) can play a more complex role by promoting the growth of a second layer of host molecules (*p*-terphenyl-3,5,3',5'-tetracarboxylic acid) above and parallel to the surface so that self-assembly is extended into the third dimension.” In the article of Blunt et al., there is an experimental demonstration that one can create a second layer by using the first underlying system, at the supramolecular level (i. e. without any covalent bond).*

Answer: We thank the reviewer for raising this point. As already pointed out by this reviewer, the multilayer C_{60} reported in Reference 1 described a case of supramolecular self-assembly based on non-covalent bonding, not on-surface synthesis (OSS). We have stated in the manuscript “Although it is straightforward to pile up molecular layers by van der Waals (vdW) interaction or hydrogen bonding [1–5], assemblies based on noncovalent bonding often have lower thermal stability, conductivity and mechanical properties compared with the covalently-bonded products

[13–15].” It is therefore important to construct the 3D covalently-bonded products precisely by OSS. Our work presents the first example to achieve this goal by enabling the interlayer covalent bonding between C₆₀ and underlying BCPM and also lateral covalent bonding between C₆₀-BCPMs.

3. In the article of Yu et al, the demonstration of the extension of 2D to 3D by the formation of covalent bonds is only suggested in supporting information (Fig. S10) and this claim is not supported by the described experimental results. Moreover, the suggested mode of 3D growth (Fig. S10) is not well described because once the BCPM molecule will have reacted with the C₆₀ by cycloaddition, it is not explained how another C₆₀ molecule will be able to react again with BCPM molecule to promote the 3D growth. Therefore, the only 3D character described in this article is justified by the 3D nature of the C₆₀ used. This result is not in accordance with the title, the abstract, the introduction and the conclusion of the article where the authors claim that their major result is “extending OSS from 2D to 3D” and that “this work unlocks an unconventional route for....3D covalently-bounded...surfaces” (lines 30-31). That claim is definitely not supported by the experimental results described in this manuscript, while it is a key point for the acceptance of an article.

Answer: We thank the reviewer for raising this point. Please refer to our responses to Comments #1 of this reviewer. Concerning the comment: *“the suggested mode of 3D growth (Fig. S10) is not well described because once the BCPM molecule will have reacted with the C₆₀ by cycloaddition, it is not explained how another C₆₀ molecule will be able to react again with BCPM molecule to promote the 3D growth”*, in fact, we have provided the requested information in the legend of the figure: *“As shown in panel a, BCPM can have cycloaddition with two C₆₀ molecules: [4+2] cycloaddition between the phenyl ring of BCPM and one C₆₀ molecule and [2+2] cycloaddition between the maleimide group of BCPM and another C₆₀ on the other side of BCPM molecule’s board.”*

4. Outside this major point, a few minor comments can also be made to the manuscript:

1) The imaging conditions (temperature, current, voltage) are missing from the captions of the figures describing the STM images. These experimental data are very important to be able to compare them with each other.

Answer: We thank the reviewer for this useful advice. All STM images presented in this work were obtained at room temperature, and we have described the details in “Experimental details” of “Methods”. Following this reviewer’s suggestion, we have added the tunneling parameters of all experimental images in the figure captions in the revised manuscript.

2) C₆₀ molecules adsorbed in the pores of the supramolecular network constituted by BCPM molecules are bright (Fig 3a-b) before the cycloadditions while they become dark (Fig 4a-b) after this reaction. Could the authors explain this difference?

Answer: We thank the reviewer for raising this question. The images in Figs. 3a-b were collected in color ranges

different from those of Figs. 4a-b. As such, the brightness/contrast of the adsorbates addressing their relative apparent heights is only meaningful when comparing the motifs/sections in the same image, not those from different images in different color ranges upon different sample preparation. In Figs. 3a–b, the guest C₆₀ molecules in the pores of the BCPM honeycomb network are evidently brighter than the surrounding BCPM molecules, because i) the top of the guest C₆₀ molecules is much higher than that of the BCPM molecules on Au(111) (9.4 Å vs. 3.4 Å); ii) the strong electron transfer from the Au substrate to these C₆₀ molecules (Fig. 3d) increases their local density of states and makes them brighter. The results shown in Figs. 4a-b are also before cycloaddition: after all the pores of the BCPM network are occupied, further C₆₀ dosage leads to each additional C₆₀ sitting on top of the phenyl ring of one BCPM ('C₆₀ on BCPM'). In the presence of BCPM underneath, the top of C₆₀ in the 'C₆₀ on BCPM' configuration (green C₆₀ in Fig. 4c) is actually much higher than the top of the guest C₆₀ (gray C₆₀ in Fig. 4c) in the BCPM pores (13.2 Å vs. 9.4 Å); as shown in Figs. 4a-b, the higher 'C₆₀s on BCPMs' present as bright spheres while the lower guest C₆₀s in the BCPM pores are much less bright.

3) Have the authors investigated the effect of thermal annealing at 370 K for 30 min on the BCPM supramolecular network without C₆₀ molecules?

Answer: We are grateful to the reviewer for asking this insightful question. We have investigated the BCPM supramolecular network on Au(111) without C₆₀ molecules annealed at 370 K for 30 min, showing no difference from that annealed at 390 K for 30 min (Fig. 2a and Supplementary Fig. 1).

4) If I understand well the discussion on the reactivity in the investigated cycloaddition (lines 207-221), the BCPM molecules act as the dienes, reacting via their HOMO, while the C₆₀ molecule plays the role of a dienophile, reacting via its LUMO (Fig S3). It seems, from Figure S9, that the addition of electron-withdrawing groups (COOH) on the BCPM molecule (diene) promotes the reaction. However, it is well known that usual [4+2] cycloadditions, which are concerted reactions, are promoted when electron-donating substituents are added to the diene and electron-withdrawing groups to the dienophile. Can the authors explain this surprising effect in more details?

Answer: We thank the reviewer for raising the question. Based on the calculated transition state, the COOH varies the reaction into an asynchronous process. The [4+2] cycloaddition requires breaking the aromaticity of the phenyl ring, where the electron-withdrawing COOH locally breaks the aromaticity of the phenyl ring by decreasing the π electron density of Carbon C and Carbon D but not Carbon A, making Carbon A more reactive (Fig. R1). Moreover, as BCPM is directly adsorbed on Au(111), although COOH is electron-withdrawing, electrons can be transferred from Au(111) to the BCPM through the tilted maleimide ring for compensation as shown in Fig. 2e.

Fig. R1 | Calculated HOMO orbitals of **a**, benzene and **b**, BCPM molecule.

Reviewer: # 2

Comments:

In this manuscript, Miao Yu et. al developed an interlayer covalent reaction by [2+4] cycloaddition between [6,6] bond of C₆₀. The morphology change upon sublimation of BCPM, C₆₀, and interlayer [2+4] cycloaddition were thoroughly characterized and has a reasonable match with DFT calculation. The resulting 3D covalently-bonded organic architectures with C_{6v} symmetry is also of interests to chemistry and organic electronics community. Overall, the synthesis method is simple while efficient, the characterization is comprehensive. I suggest to be published in nature communication after these comments have be addressed:

Answer: We are grateful to the reviewer for taking the time to evaluate our manuscript and providing positive remarks and constructive comments. Please refer to our point-by-point responses below.

1. *Fullerene derivatives are known to have poor thermostability so that fewer materials have been used for vacuum-deposition process. Please show the TGA result of pristine C₆₀, BCPM, and resulting C₆₀-BCPM to show the thermostability at 490 K, at which temperature authors checked the reactivity of C₆₀-BCPM.*

Answer: We appreciate the reviewer for offering this important advice. Although some fullerene derivatives have relatively-poor thermal stability, C₆₀ molecules are known for its high stability [e.g. *Science* **271**, 317-324 (1996)]; thermal evaporation of C₆₀ for vacuum deposition has been widely applied [e.g. *Surf. Sci.* **295**, 13-33 (1993); *Phys. Rev. Lett.* **99**, 226105 (2007); *Phys. Rev. Lett.* **103**, 056101 (2009); *Phys. Rev. Lett.* **71**, 2959 (1993); *J. Chem. Phys.* **134**, 044707 (2011)]. BCPM is also stable upon the thermal evaporation as demonstrated by the intact morphology of the molecules in the STM images, which are also consistent with our earlier results on surfaces [*Nat. Chem.* **12**, 1035–1041 (2020); *Angew. Chem., Int. Ed.* **60**, 17435–17439 (2021)]. As C₆₀-BCPM is fabricated using on-surface synthesis (OSS), it is rather challenging to extract it from Au(111) as well as to obtain a sufficient amount of C₆₀-BCPM for TGA. This is a common issue for almost all products obtained from OSS. Still, as demonstrated in Fig. 6,

the cycloaddition between C_{60} and BCPM forming C_{60} -BCPM occurs at 490 K, indicating that C_{60} -BCPM is thermally stable at this temperature.

2. To prove the novelty of the 3D structure formed by interlayer covalent, please also attach the test result of direct sublimation of C_{60} -BCPM onto Au(111) surface.

Answer: We thank the reviewer for providing this useful suggestion. As a matter of fact, we only synthesized C_{60} -BCPM on Au(111) in ultrahigh vacuum conditions by OSS. In solution synthesis, it is not easy to selectively synthesize C_{60} -BCPM compound considering the multiple possibilities of covalent bonding between C_{60} and BCPM. It is, thus, not practical to obtain pure C_{60} -BCPM compound and directly sublime the C_{60} -BCPM compound onto the surface. Moreover, lacking the confinement of the surface template in our step-by-step deposition, even if there was C_{60} -BCPM compound synthesized other than OSS, when deposited on Au(111), their arrangement and degree of order on the surface would be much lower than those presented in the present work: additional adsorption geometries of C_{60} -BCPM would be allowed when depositing C_{60} -BCPM compound, depending on competition between the strong affinity of C_{60} end and BCPM end with Au(111), the molecular coverage, the dosing rate and substrate temperature.

Reviewer: # 3

Comments:

The present manuscript reports the on-surface synthesis (OSS) of 3D molecular architectures (i.e., structures with more than one molecular layer in height) by exploiting the cycloaddition reaction between C_{60} and BCPM at the surface of Au(111). The paper is concise and well-structured. The results are generally well discussed, and the authors try to explain the experimental observations. The development of surface built-in 3D molecular architectures represents a significant advancement in surface science. I consider this work an important contribution to the field, although I believe it can still be improved in its scientific content and presentation. Although the experimental results constitute good proof of what is happening at the surface, I am skeptical about some interpretations of the phenomena.

Answer: We are grateful to the reviewer for taking the time to evaluate our manuscript and providing positive remarks, constructive comments and thought-provoking ideas. Following the reviewer's suggestions, we have carried out additional DFT calculations and provided more analysis/explanations in the revised manuscript/Supporting Information. Please refer to our point-by-point responses below.

1. The bonding between C₆₀ and BCPM on the Au(111) surface

1) Some details of the theoretical study of the [4+2] reaction need some clarification. Firstly, the authors admit that the reaction involves the HOMO of the diene and the LUMO of the dienophile. According to Supp. Fig. 3, however, the situation is not that clear, because there is a smaller energy gap between the LUMO of the diene and the HOMO of the dienophile. This is consistent with the presence of the π -electron withdrawing -COOH groups in the diene. In these cases, an inverse Diels-Alder cycloaddition can occur, involving the LUMO of the diene and the HOMO of the dienophile. The authors should clarify this point.

Answer: We are grateful to the reviewer for raising these important comments. The Diels-Alder cycloaddition not only requires a relatively small gap between the LUMO of the diene and the HOMO of the dienophile but also requires the appropriate steric configuration allowing the wave functions of the LUMO and HOMO overlapped with each other. Following this reviewer's suggestion, we have performed additional calculations. The results have been added in new Supplementary Fig. 3 and discussed in the revised Supplementary Information (Page 4) as follows:

“As shown in Supplementary Fig. 3, the HOMO of BCPM is essentially located at the phenyl ring, and its LUMO is localized around the maleimide ring. As a typical dienophile, C₆₀ has a relatively low-lying LUMO and its [6,6] bonds can act as preferential sites for [4+2] cycloaddition [2–4]. Upon adsorption of C₆₀ on the BCPM layer, after the pores of BCPM layer are occupied by the guest C₆₀ molecules, each excess C₆₀ molecule sits exactly on top of one BCPM's phenyl ring whilst the mobility of C₆₀ moving to the top of the maleimide ring is constrained (Fig. 4c). The steric geometry between C₆₀ and BCPM allows the overlap hence hybridization only between the HOMO of BCPM and the LUMO of C₆₀ (Supplementary Figs. 3c and 3g). The energy difference between these molecular orbitals is 3.78 eV, which is suitable to form a bonding orbital for cycloaddition [5]. Moreover, the wave function of the LUMO of BCPM and HOMO of C₆₀ also show a good match in the orbital symmetry for hybridization. As there is no overlap between the LUMO of BCPM and the HOMO of C₆₀ due to this steric hindrance (Supplementary Figs. 3f), their inverse [4+2] cycloaddition cannot take place.”

Supplementary Fig. 3 | **a–c**, Calculated HOMO of BCPM, LUMO of C₆₀ and the hybridization between the HOMO of BCPM and the LUMO of C₆₀ when C₆₀ sitting on the phenyl ring of BCPM. **d–f**, Calculated LUMO of BCPM, HOMO of C₆₀ and the non-overlapped LUMO of BCPM and HOMO of C₆₀ in the given steric configuration. **g**, Energy level diagram of the LUMO and HOMO of BCPM and C₆₀ together with the bonding orbital and anti-bonding orbital of their adduct, i.e. C₆₀-BCPM.

2) My second concern are the calculated energies for the [4+2] reactions. Cycloaddition reactions are known to be substantially exothermic; however, the results depicted in Supp. Figs. 8a and 9 suggest otherwise (the calculated reactions are highly endothermic, around +90 kJ/mol). In the light of these results, one may ask what is the driving force for this reaction (it is enthalpically and entropically disfavored...)? Is there any problem with the calculations? The DFT-D3 method seems adequate for these systems with considerable dispersion forces. Have the authors considered the correction for the basis-set superposition error (BSSE)? For example, the initial states in Supp. Figs. 8a and 9 can be erroneously stabilized by BSSE due to the formation of the van der Waals complex between the diene and dienophile. It is true that in all systems considered in Supp. Fig. 9 (including BCPM) there is a significant breaking of aromaticity in the diene upon cycloaddition, which contributes for making the reaction less exothermic. With all these concerns, I'm not sure if the reaction occurring at the surface between C₆₀ and BCPM is the [4+2] cycloaddition proposed by the authors. I'm not an expert of STM, but I think this technique cannot provide many details about the nature of the bonding occurring at the surface. The location and structure of the molecules above the surface are clear, however, the way the molecules bond among each other is open to speculation. Given this, I ask if there are other bonding models compatible with the experimental observations? The authors should review the calculations to check for eventual errors in the calculated energies. Additionally, they should provide more experimental evidence for the type of reaction occurring between C₆₀ and BCPM. For instance, they could make a reaction control test between C₆₀ and BCPM in solution and characterize the product. Has this been done by other authors before?

Answer: We thank the reviewer for asking questions about the calculations, as it gives us the opportunity to clarify

certain important aspects. We used Vienna *ab initio* simulation package (VASP) to perform the calculations in which plane-wave basis was used and BSSE was thus not required. Following this reviewer's suggestion, we have double checked all the calculations.

The cycloaddition reaction in the present system is endothermic but raises no concern, as it is well consistent with various cases reported previously where the cycloadditions showed a similar endothermic process [e.g. *Chem. Eur. J.* **25**, 9902–9912 (2019)] and other on-surface reactions showed even higher reaction energies and energy barriers [e.g. *J. Am. Chem. Soc.* **138**, 2809-2814 (2016); *Angew. Chem., Int. Ed.* **55**, 9881-9885 (2016); *J. Phys. Chem. C* **118**, 3181-3187 (2014)].

All the models and reaction processes we used were based on the experimental observations. The confirmation for the occurrence of [4+2] cycloaddition between C₆₀ and BCPM does not rely on the observation of direct bonding of them using STM given that BCPM is beneath C₆₀ in the present case, but rather on the following evidences: i) C₆₀-BCPM shows the triangle-shaped submolecular structure of C₆₀ at room temperature (RT) (Figs. 5c and 5e), whilst pristine C₆₀ shows no submolecular contrast due to its rotation at RT; as C₆₀ molecules are bonded to BCPM with their bottom [6,6] bonds, their opposite [6,6] bonds on the top thus can be observed in triangular morphology; ii) C₆₀-BCPM is 3.4 Å higher compared to pristine C₆₀ on Au(111) (Fig. 5d); iii) further intermolecular covalent bonding forming the macrocycles also only occurs for C₆₀-BCPM instead of pristine C₆₀; and iv) BCPM in C₆₀-BCPM remained on the surface well above the desorption temperature of pristine BCPM (490 K vs. 450 K).

The C₆₀-BCPM bonding model we presented here is the only reasonable one, given that i) the triangular submolecular feature is the only intramolecular structure observed from C₆₀-BCPM, ii) the products on the surface are uniform, iii) the height of C₆₀-BCPM observed is consistent with the [4+2] cycloaddition with C₆₀ on the phenyl ring of BCPM, iv) the spatial hindrance rules out an alternative position for the overlap hence hybridization between C₆₀ and BCPM.

In solution synthesis, without surface confinement, it is difficult to selectively synthesize the C₆₀-BCPM compound presented in this work, as there are multiple possibilities of covalent bonding between C₆₀ and BCPM in solution, e.g. C₆₀ bonded on the maleimide ring of BCPM, two C₆₀ molecules bonded on one BCPM, two/more BCMPs bonded on one C₆₀, etc.

2. The (im)possibility of a thermal [2+2] cycloaddition

1) Although the experimental observations give strong support for the formation of macrocycle hexamer rings after annealing at 490 K, I have some doubts about the provided rationalization. The authors state that the various C₆₀ in the hexamer are covalently bonded via [2+2] cycloaddition. However, [2+2] cycloadditions are generally thermally forbidden and need to be photochemically induced to occur (with rare exceptions). In the conditions reported by the authors I am not convinced that the C₆₀ molecules in the hexamer react via [2+2] cycloaddition. Actually, solid C₆₀ is

stable at 490 K, and no covalent bonding between the various molecules in the crystal is observed. Upon heating to 490 K the molecules in the crystal remain covalently separated, like in a regular crystalline solid.

Answer: We are grateful to the reviewer for raising this point. In fact, [2+2] cycloaddition between C_{60} can be photochemically induced, yet can also be triggered without irradiation, e.g. by thermal excitation and high pressure [*Carbon* **36**, 319–343 (1998); *Carbon* **82**, 381–407 (2015)]. We have calculated the transition state of [2+2] cycloaddition between two C_{60} -BCPM molecules on Au(111) (Supplementary Fig. 10b), deducing an activation energy of 1.87 eV, which can be easily conquered by a thermal excitation in OSS [*J. Am. Chem. Soc.* **138**, 2809-2814 (2016); *Angew. Chem., Int. Ed.* **61**, 5 (2022); *J. Am. Chem. Soc.* **133**, 38 (2011); *J. Am. Chem. Soc.* **142**, 31 (2020); *Nat. commun.* **7**, 11002 (2016)]. Following the reviewer's suggestion, we have carried out additional calculations. The HOMO-LUMO gap of C_{60} -BCPM is found to be 0.23 eV smaller than that of C_{60} (2.54 eV vs. 2.77 eV). As [2+2] cycloaddition between C_{60} molecules (or C_{60} part of C_{60} -BCPM) involves both the LUMO and HOMO, the smaller gap would facilitate the cycloaddition. Moreover, the calculated energy barrier of [2+2] cycloaddition (in gas phase) between the C_{60} parts of C_{60} -BCPMs is also 0.16 eV smaller than that of C_{60} (1.77 eV vs. 1.93 eV). In addition, the forbidden rotation and appropriate orientation of C_{60} bonded with BCPM on Au(111) may also promote the reaction rate within a given duration at a given temperature. we have added the results in new Supplementary Fig. 12 and discussed accordingly in the revised manuscript (Page 11) as follows:

“According to DFT calculations, the HOMO-LUMO gap of C_{60} -BCPM is 0.23 eV smaller than that of C_{60} and the energy barrier of [2+2] cycloaddition (in gas phase) for C_{60} -BCPM is also 0.16 eV lower than that of C_{60} . In this regard, [2+2] cycloaddition among C_{60} parts of C_{60} -BCPMs is promoted compared with the case of pristine C_{60} (Supplementary Fig. 12).”

Supplementary Fig. 12 | Initial state (IS), transition state (TS) and final state (FS) geometry and the energy diagram for [2+2] cycloaddition between **a**, two C_{60} -BCPMs and **b**, two C_{60} s.

2) The STM images are consistent with the presence of the hexamer but furnish little information about the bonding nature among the C_{60} s. There are other possibilities for explaining this bonding, as for example single, or sp^2 -like bonds between the C_{60} .

Answer: We thank the reviewer for providing this useful advice. Following this reviewer's suggestion, we have added the following structural models in new Supplementary Fig. 8 and described it in the revised manuscript (Page 10) as follows:

"We then analyzed the bonding among C_{60} -BCPMs within the HR by DFT calculations. The possibility of single bond between C_{60} -BCPMs is ruled out: when a single bond is set in the initial structures of two C_{60} -BCPMs or the HR, the second bond forms after optimization, indicating that single bond between C_{60} -BCPMs is not stable (Supplementary Fig. 8). The optimized structure reveals that the geometry of C_{60} -BCPMs in the HR accepts [2+2] cycloaddition between [6,6] side bonds of C_{60} part (Supplementary Fig. 8d and Fig. 6e), consistent with C_{60} polymerization induced at high temperature and pressure [40,41]."

sp^2 -like bond between C_{60} s is not considered: the [6,6] bond (double bond) is more reactive than the other bonds of C_{60} , and it forms sp^3 in the adduct of C_{60} .

Supplementary Fig. 8 | **a**, Initial structure, and **b**, optimized structure by DFT calculations for bonding between two C_{60} -BCPMs. When initially setting a single bond between them, the second bond forms after full relaxation. **c**, Initial structure, and **d**, optimized structure for bonding between the adjacent C_{60} -BCPMs in the HR. Similar to the case of two C_{60} -BCPMs, when we set a single bond between the adjacent C_{60} -BCPMs, the final structure after optimization shows two bonds between adjacent C_{60} -BCPMs.

3) Alternatively, and perhaps more probable, the hexamers are just the parent structures of C_{60} sitting above the original BCPM flower structure. Heating at 490 K shall provide enough thermal energy for the C_{60} molecules to acquire significant surface mobility and become more scarce and less structured. The resulting hexamers are possibly just the remaining structures that survived heating.

Answer: We are grateful to the reviewer for raising this comment. We can rule out the possibility that "the hexamers are just the parent structures of C_{60} sitting above the original BCPM flower structure" based on the following evidences: i) after annealing the C_{60} layer on BCPM at 370 K for 30 min, the initial network of BCPM flowers has vanished, while two alternative close-packed structures, *i.e.*, 'D1' and 'D2', emerged (Fig. 5); ii) annealing at 370 K and then 490 K, no noticeable molecular desorption is observed upon the annealing process; iii) the experimental intermolecular distance of C_{60} -BCPMs in the HR is distinctly smaller than the distance between C_{60} s on the BCPM flowers (8.4 Å vs. 10.2 Å). Following this reviewer's suggestion, we have anyway further calculated the bonding between C_{60} s when they are sitting above the BCPM flower at different distance. According to DFT calculations, the

C_{60} - C_{60} covalent bonds only form when their distance is $< 8.9 \text{ \AA}$; when the distance is $\geq 8.9 \text{ \AA}$, the two C_{60} molecules will move away from each other to 10.0 \AA (a distance close to the intermolecular distance in the C_{60} crystal with vdW interaction) (Fig. R2). Therefore, the experimentally measured intermolecular distance of 8.4 \AA in HR directly indicates the covalent bonding between the C_{60} parts. If the hexamers are just the remaining parent structures of C_{60} sitting above the original BCPM flower structure without covalent bonding, it would be easily disrupted during the STM scanning when the tip is really close to the sample (Supplementary Fig. 7).

Fig. R2 | a, Initial structure and **b**, final structure of two C_{60} molecules on BCPM flower with an initial distance of 8.9 \AA . **c**, Initial structure and **d**, final structure of two C_{60} with an initial distance of 9.0 \AA .

4) In page 9, line 175 the authors say that “the sample was annealed at 490 K for 30 min ” but do not give important details about this sample. I presume that it is a sample like the one depicted in Fig. 5 a (after annealing the C_{60} -BCPM network at 370 K). In that case, the observed hexamers after heating at 490 K can be the result of a D1 territory dispersing through a D2-like one (or similar).

Answer: We thank the reviewer for this insightful comment. The same sample was treated in a successive process, *i.e.* annealed at 370 K then annealed at 490 K . Following this reviewer’s suggestion, we have added the following discussion in the revised manuscript (Page 9):

“Linescans show that the HR’s diameter is 16.8 \AA (Fig. 6d); the center-to-center distance between two neighboring protrusions in an HR is only 8.4 \AA , which is largely reduced compared with that of C_{60} -BCPM in D1 before annealing at 490 K . The largely reduced spacing suggests that C_{60} -BCPMs in this case do not interact by vdW as would be the case of isolated molecules, but covalently bond with one another. Moreover, the height difference between the HR and the less bright protrusions (Figs. 6a and 6b) is much less significant than the height difference between D1 and D2 domains. Both the small spacing of C_{60} -BCPMs and height difference exclude the possibility of forming the HRs by mixing D1 with D2.”

5) In page 9, lines 178-182 the authors do refer that the hexamers have a smaller radius than the analogous structures observed in D1 motifs. However, without experimental uncertainties one cannot judge if this difference is statistically significant.

Answer: We thank the reviewer for raising this concern. The distance between the neighbouring protrusions of the hexamer is 8.4 Å, which is 1.7 Å smaller than the periodicity of the unreacted *D1* (10.2 Å). The difference of 1.8 Å is sufficiently significant and cannot be induced by the experimental uncertainties. The experimental uncertainty is less than 5% determined by scanning of pristine Au(111) with atomic resolution.

6) Also, the authors are comparing the diameter of the C_{60} hexamers with the layer below (Fig. 6 b and d), which I'm not sure what it is... Is it Au, the network of BCPM or the similar C_{60} -BCPM layer observed in *D1*? Note that the position of the hexamers above the background layer does not seem very plausible with the presence of a *D1* structure below or side-by-side. Comparisons should be made between the C_{60} hexamer rings (Fig.6 b) with those similar structures observed in *D1* motifs (Fig. 5 c). Although the authors say it correctly in the Fig. 6 caption (for panel d), the images are confusing and I'm not sure if the green arrow corresponds to a *D1* motif. The authors should clarify this point and describe in more detail the resulting surface after annealing at 490 K.

Answer: The periodicity of the domain marked by the green arrow in Fig. 6 is consistent with that of *D1* before annealing at 490 K. Besides, the apparent height difference between the less bright protrusions in the domain with the HRs is quite small (less than 1 Å) and thus can be only attributed to that of close-packed C_{60} -BCPMs and lateral covalent bonded C_{60} -BCPMs which have similar geometrical heights, not mixture of C_{60} s and C_{60} -BCPMs as C_{60} -BCPM is 3.5 Å higher than C_{60} on Au(111). The comparison between HR and the less bright protrusions in the domain was made in the same image to avoid the tip influence under different tunneling conditions. We have compared HRs (Fig. 6b) with the unreacted *D1* domain (Fig. 5c) in the manuscript (Page 9) as follows:

*"Linescans show that the HR's diameter is 16.8 Å (Fig. 6d); the center-to-center distance between two neighboring protrusions in an HR is only 8.4 Å, which is largely reduced compared with that of C_{60} -BCPM in *D1* before annealing at 490 K."*

For both the pristine *D1* (before lateral [2+2] cycloaddition) and the domain showing HRs (after lateral cycloaddition), the single layer of C_{60} -BCPMs adsorbs directly on the Au(111) surface. Following this reviewer's suggestion, we have added a more detailed description in the revised manuscript (Page 9) as follows:

*"To explore the reactivity of C_{60} -BCPM, the sample (as presented in Fig. 5) was further annealed at 490 K for 30 min. *D1* remains the close-packed structure. Considering the low diffusion barrier of C_{60} -BCPM on Au(111) (Supplementary Fig. 5), the well-maintained lateral arrangement is attributed to the intermolecular interaction among C_{60} -BCPMs. Interestingly, bright macrocycles emerge in *D1*, where the hexamer rings (HRs) are dominant (Fig. 6a and Supplementary Fig. 6)."*

7) Even if these diameters are proven to be different, that does not prove conclusively the existence of covalent bonding among the C_{60} s in the hexamers. The smaller diameter in the hexamers (Fig. 6 b) can be due to other factors, like the absence of neighboring fullerenes or of the central physisorbed C_{60} (the grey C_{60} in Fig. 4). If a [2+2]

cycloaddition occurs the authors should provide more convincing experimental proofs.

Answer: We thank the reviewer for this useful comment. Based on DFT calculations, after structural optimization, the intermolecular distance between two covalently-bonded C₆₀-BCPMs is 8.1 Å, whilst the distance for neighbouring C₆₀-BCPMs interacted with vdW interaction is 9.8 Å. Therefore, the short intermolecular distance of C₆₀-BCPMs (8.4 Å) is a direct evidence of the covalent bonding among C₆₀-BCPMs in the hexamer, which cannot be formed by C₆₀-BCPMs with vdW interactions (9.8 Å). Moreover, referring to our responses to Comments #3, single bond between C₆₀-BCPMs in the HR is not stable nor fits the experimental results; only the double bond of [2+2] cycloaddition between C₆₀ [6,6] side bonds matches the experimental observations.

8) The authors should also provide more details about the sample subjected to annealing at 490 K and prove (or disprove) the hypothesis of thermal diffusion through the surface.

Answer: We thank the reviewer for raising the important comments. Following this reviewer's suggestion, we have calculated the diffusion barrier for a single BCPM, C₆₀, and C₆₀-BCPM on Au(111), respectively, using cNEB calculations to search the minimum energy path (MEP) for a diffusion length of 5.08 Å (for the top-to-top sites as marked by the black arrow in Fig. 2a) along the [11 $\bar{2}$] direction of Au(111). The resulting diffusion barriers for individual BCPM, C₆₀ and C₆₀-BCPM on Au(111) are 0.01 eV, 0.09 eV and 0.06 eV, respectively, which are rather small and benefit the ordered assembly of BCPM, C₆₀ and C₆₀-BCPM upon annealing. We have added the results in Supplementary Fig. 5 and described them in the revised manuscript (Page 9) as follows:

“To explore the reactivity of C₆₀-BCPM, the sample (as presented in Fig. 5) was further annealed at 490 K for 30 min. D1 remains the close-packed structure. Considering the low diffusion barrier of C₆₀-BCPM on Au(111) (Supplementary Fig. 5), the well-maintained lateral arrangement of D1 is attributed to the intermolecular interaction among C₆₀-BCPMs.”

Supplementary Fig. 5 | a, The model of Au(111) surface, where the black arrow shows the diffusion path of the molecule for the calculation of minimum energy path (MEP) profiles. The calculated MEP profile for **b**, a single BCPM, **c**, a single C₆₀ and **d**, a single C₆₀-

BCPM on Au(111).

3. Uncertainties associated to STM measurements. *The authors should indicate the calculated uncertainties (if not possible, an educated estimation), associated to the distance measurements by STM.*

Answer: We thank the reviewer for raising this important point. The experimental uncertainty is less than 5% determined by scanning of pristine Au(111) with atomic resolution.

4. Minor comments *(following these comments shall improve the quality of the manuscript in its presentation and exactness)*

1) The use of the notation “[2+4] cycloaddition”. *All through the manuscript, the authors refer to the [4+2] cycloaddition reactions as [2+4]. Although this is not incorrect, the accepted notation is [4+2] (4 π electrons from the diene and 2 π electrons from the dienophile). Please correct the notation.*

Answer: We thank the reviewer for this useful advice. Following this reviewer’s suggestion, we have modified [2+4] cycloaddition to [4+2] in the revised manuscript.

2) Page 3, lines 57-60, the concept of a 3D molecule. *The authors speak of the concept of a 3D molecule. What is exactly meant by a “3D molecule”? Why is C_{60} the most representative “3D molecule”? Every molecule can be considered 3D... The authors also refer “unique 3D conjugated electronic structure”. The use of the term “3D” in these contexts is confusing. Please explain, or just remove this.*

Answer: We are grateful to the reviewer for this helpful suggestion. Accordingly, we have modified “3D molecule” and “3D conjugated electronic structure” into “**the most representative fullerene**” and “**unique conjugated electronic structure**” in the revised manuscript, respectively.

3) Page 3, line 57, properties of C_{60} . *Saying that “ C_{60} has magnificent physical properties/potentials” is misleading, because, for example, C_{60} as a single molecule does not present superconductivity. Some properties are specific to a particular state of matter, not necessarily inherent to the molecule. Moreover, many times such magnificent properties are reported for functionalized C_{60} , not C_{60} itself. This sentence, as it is, is a bit vague. Please be more specific of which state/form of C_{60} display the properties referred.*

Answer: We thank the reviewer for this useful advice. Accordingly, we have rephrased the revised manuscript (Page 3) as follows:

“As the most representative fullerene, C_{60} and its derivatives have shown magnificent physical properties/potentials (e.g. pressure resistance of solid C_{60} [16], optical restriction of C_{60} solution [17], superconductivity of alkali-metal doped C_{60} crystal [18], magnetism of TDAE- C_{60} compound [19]).”

4) Page 3, lines 65-66, and in more instances of the manuscript – the idea of spatial requirements for the reaction with C₆₀. In some parts of the manuscript (page 3 line 65; page 3 line 69; page 4 line 84; page 8 line 163; page 12 line 239), the authors imply that C₆₀ and BCPM have an appropriate steric configuration for the reaction. This is not surprising given the spherical symmetry of C₆₀ and the fact that BCPM is immobilized at the surface. Hence, this reaction shall not be particularly characterized by orientation requirements of the reacting molecules, other than the well-known spatial requirements of the overlapping orbitals in the diene and dienophile in [4+2] cycloadditions. In my opinion, the idea that should be conveyed here is that adsorbed BCPM has a geometry that allows it to act like a diene in a subsequent reaction with C₆₀. On the other side, due to the high symmetry of C₆₀ (every [6,6] bond is equally reactive), there are actually no significant orientational requirements for it to react with BCPM. I understand what the authors mean, but, as it is, it may pass the idea that C₆₀ must approach BCPM with a very specific orientation. Given the nature of the system under study (immobilized BCPM and symmetrical C₆₀) it is easily recognized that spatial requirements will be a minor issue for the reaction; there is no need to emphasize it so much in the manuscript.

Answer: We thank the reviewer for this interesting discussion. We emphasize the steric configuration for the interlayer cycloaddition between BCPM and C₆₀ because it is distinct from the conventional on-surface synthesis where the precursors in the reported cases are in the same layer and the reaction on the substrate occurs laterally. In our experiments, the guest C₆₀ molecules in the pores of BCPM network (Fig. 3) cannot have cycloaddition with the BCPM molecules even upon annealing. Only when one C₆₀ sits above the phenyl ring of one BCPM, [4+2] cycloaddition between BCPM and C₆₀ is enabled which requires the [6,6] bond of C₆₀ facing the phenyl ring of BCPM.

5) Page 11, lines 221-223. Where the authors say “We believe that this is related to the rotation of pristine C₆₀ molecules at RT (or a higher temperature) [38,39], making the covalent coupling between neighbouring C₆₀ impossible. ” Rotation of a molecule does not prohibit the formation of covalent bonds. If this was the case, reactions in solution, liquid and gas phase would be impossible. Please correct it.

Answer: We are grateful to the reviewer for offering this advice. Following this reviewer’s suggestion, we have removed the sentence in the revised manuscript.

6) Page 11, lines 224-233, the possibility of growing the 3D motif. The discussion of further evolving the 3D motif of fullerene-BCPM layers above the Au surface is very interesting and relevant for the discussion. The authors correctly recognize that this is probably experimentally challenging and refer the less surface-confinement effect and higher freedom of molecules. However, I see one important difficulty for this task that should also be mentioned here: the possibility that reactions on C₆₀ after the first coupling with BCPM (or any other addend) can produce many possible isomers, many of which will not have an adequate orientation to grow the 3D structure. The authors only consider the more obvious trans-1 isomer (Supp Fig. 10 b), but many more are possible. Achieving high isomer

selectivity in these syntheses is not an easy task.

Answer: We thank the reviewer for raising this concern. Following this reviewer's suggestion, we have added the following sentence in the revised manuscript (Page 12):

*"Considering the **reduced** surface-confinement effect hence higher freedom of the molecules **as well as the possible unfavorable orientations of the formed isomers**, the experimental realization may be challenging, requiring fine control and further exploration."*

7) Small typos and imprecisions along the manuscript

- Page 3, line 62, change "denes" to "dienes".

- Page 3, line 75, change "forming macrocycles with the hexamers rings with C_{6v} symmetry Dominated" to "forming macrocycles mostly consisting of hexamer rings with C_{6v} symmetry".

- Page 7, line 142, change "two types of territories is observed" to "two types of territories are observed".

- Page 8, line 156, where the authors say "showing the intramolecular structure of C₆₀" did they mean intermolecular? Intramolecular implies covalent bonding among the molecules. The term "intermolecular" is more correct in this context, since it describes the organization of various C₆₀ molecules interacting via intermolecular interactions, like in the crystal phase.

Answer: We thank the reviewer for pointing out these errors. Following this reviewer's suggestions, we have made the corrections in the revised manuscript accordingly for Line 62 on Page 2, Line 75 on Page 3, Line 142 on Page 7.

For "intramolecular structure" in Line 156 on Page 8, we have changed it to "**triangular submolecular feature**". We emphasize the distinct triangular intramolecular feature of C₆₀.

Reviewers' Comments:

Reviewer #1:

Remarks to the Author:

On the basis of my review (see attached file), I recommend the rejection of this article because experimental proof of 3D growth is lacking.

I thank the authors for the rebuttal letter.

As I mentioned in my first review, my major point is: do the experimental results described in this article actually demonstrate the proposed title “Extending on-surface synthesis from 2D to 3D by cycloaddition with C₆₀”?

The answer of the authors is: “The point of 3D synthesis is demonstrated experimentally based on both [4+2] cycloaddition between C₆₀ and BCPM and [2+2] cycloadditions of C₆₀-BCPMs. In this way, both vertical and lateral covalent coupling are enabled in this system, presenting a prototype of 3D synthesis. In addition, we have investigated the double cycloaddition between C₆₀ and BCPM by DFT calculations and described/discussed the results in both main text (Page 12) and Supporting Information (Supplementary Fig.13). The results confirm that BCPM and C₆₀ can form multiple layers of covalently-bonded BCPM-C₆₀-BCPM-C₆₀, where the phenyl ring and maleimide ring of BCPM can provide double sites for cycloaddition with C₆₀, and the multiple [6,6] bonds of C₆₀ allows it to react with BCPM at both the top and bottom [6,6] bonds by cycloaddition. Following this pattern, **3D covalently-bonded organic architectures/devices with extended/controlled thickness could be fabricated bottom-up via cycloaddition reaction like Lego pieces with two connectors.**”

The extension to 3D is only based on the DFT calculations of a single C₆₀-BCPM-C₆₀ or a single BCPM-C₆₀-BCPM in the vacuum (see below) while it is called “interlayer cycloaddition”

13. Formation of C₆₀-BCPM-C₆₀ and BCPM-C₆₀-BCPM by interlayer cycloaddition

Supplementary Fig. 13 | The DFT-optimized structure of **a**, C₆₀-BCPM-C₆₀ and **b**, BCPM-C₆₀-BCPM. As shown in panel a, BCPM can have cycloaddition with two C₆₀ molecules: [4+2] cycloaddition between the phenyl ring of BCPM and one C₆₀ molecule and [2+2] cycloaddition between the maleimide group of BCPM and another C₆₀ on the other side of BCPM molecule's board. As shown in panel b, one C₆₀ molecule can have cycloaddition with two BCPM molecules, one on the top and the other at the bottom.

As it is claimed in the manuscript (p5, top) the underlying surface and the molecule-molecule play a role in the formation of the BCPM network. Then, the growth of the network of fullerenes is also driven by the underlying BCPM layer (p7, top). Due to these claims, the extension toward 3D is not straightforward and it cannot be simulated by only three molecules in the vacuum.

To sum up, I'm still not convinced by this answer. After a very fine (re-)reading of this article, the transition to 3D is still only a hypothesis based on numerical simulations (without underlying surface, without self-assemblies) and the use of the conditional mode by the authors confirms this point of view. If the main scientific issue of this article is the extension towards the 3rd dimension (as indicated in the title and the introduction) then I think that the authors should give an experimental proof of this 3D extension.

Reviewer #2:

Remarks to the Author:

The authors have responded all the comments I gave with reasonable reply, thus I would recommend this paper to be published in Nature Communication

Reviewer #3:

Remarks to the Author:

The authors made significant changes to the manuscript, in which they addressed most of my concerns and of the other reviewers. The revised version of the manuscript is clearer and scientifically more rigorous. Since I'm not an expert in the field of on-surface synthesis I cannot evaluate the immediate impact or application of this paper in the respective scientific community. However, in my opinion, this work is a beautiful example of molecular and supramolecular chemistry at the surface, and merits publication in Nature Communications.

The experimental methodology was well-planned, and the results seem robust and consistent. The sequential experimental characterization of the surface is for me the strongest point of this work. I'm still a bit skeptical about the computational results. My biggest concern is that according to the DFT results, there is no significant driving force for the [4+2] cycloaddition reaction (it is highly endothermic and, in principle, not favorable in entropic terms either). However, it appears to occur in appreciable extent at the Au surface and be rather stable, considering the low reverse barrier of about 20 kJ/mol. There is my general opinion that DFT is not that accurate for relatively complex systems. There are many electronic effects that, being erroneously accounted for by the computational method can easily scale-up to significant errors. Many times, obtaining accurate computational energies requires a careful balance of effects (like for instance, the use of homodesmotic schemes for canceling effects in reactants and products). Given this, in my opinion, the interpretation of the experimental STM results is excessively anchored on the DFT results. I would appreciate more experimental work to validate some of the claims, for example (a) TGA of a solid C60-BCPM sample to evaluate decomposition temperature; (b) characterization of the hexamers after desorption of the surface (e.g. by washing). I understand, however, that carrying out these studies is not trivial and that the paper already reports a good amount of valid experimental data to justify publication as a communication.

Some minor corrections:

- Page 3, correct "comound" to "compound" in "...magnetism of TDAE-C60 comound [19]"
- Page 12, top, where it says "C60-BCPM is also lower than that of C60 (Supplementary Fig. 12)", referring to the calculated energy barrier to [2+2] cycloaddition; according to Supplementary Fig. 12 this barrier is lower to C60 (1.77 eV compared to 1.93 eV to C60-BCPMs). Hence, the following conclusion "In this regard, [2+2] cycloaddition among C60 parts of C60-BCPMs is promoted compared with the case of pristine C60" does not apply. Were the authors referring to the energy of the reaction, instead of the energy barrier? Please correct.

Reviewer: #1

Comments:

I thank the authors for the rebuttal letter.

Answer: We are grateful to the reviewer for taking the time to evaluate our revision and raising the following concern.

1. As I mentioned in my first review, my major point is: do the experimental results described in this article actually demonstrate the proposed title "Extending on-surface synthesis from 2D to 3D by cycloaddition with C₆₀"? The answer of the authors is: "The point of 3D synthesis is demonstrated experimentally based on both [4+2] cycloaddition between C₆₀ and BCPM and [2+2] cycloadditions of C₆₀-BCPMs. In this way, both vertical and lateral covalent coupling are enabled in this system, presenting a prototype of 3D synthesis. In addition, we have investigated the double cycloaddition between C₆₀ and BCPM by DFT calculations and described/discussed the results in both main text (Page 12) and Supporting Information (Supplementary Fig.13). The results confirm that BCPM and C₆₀ can form multiple layers of covalently bonded BCPM-C₆₀-BCPM-C₆₀, where the phenyl ring and maleimide ring of BCPM can provide double sites for cycloaddition with C₆₀, and the multiple [6,6] bonds of C₆₀ allows it to react with BCPM at both the top and bottom [6,6] bonds by cycloaddition. Following this pattern, 3D covalently-bonded organic architectures/devices with extended/controlled thickness could be fabricated bottom-up via cycloaddition reaction like Lego pieces with two connectors." The extension to 3D is only based on the DFT calculations of a single C₆₀-BCPM-C₆₀ or a single BCPM-C₆₀-BCPM in the vacuum (see below) while it is called "interlayer cycloaddition"

Answer: We realize that the reviewer has confused the concept of 3D on-surface synthesis (OSS) with growth of multiple molecular layers on surfaces. Quoted above by this reviewer, the 3D OSS we referred to in this work is the non-planar multiple cycloaddition reactions of C₆₀ in the presence of BCPM, where C₆₀ acts as the "multi-plug" reactant enabling the vertical covalent coupling between the C₆₀ and BCPM molecular layers perpendicular to the surface and the lateral covalent bonding within the C₆₀-BCPM molecular layer parallel to the surface. This is distinct from all the OSS cases reported previously (e.g. Ullmann, Glaser coupling), which are based on the dissociation/formation of organic precursors' σ bonds at the terminal of molecular precursors and can only be achieved by lateral coupling on surfaces, thus confining OSS to be 2D.

Inspired by this reviewer, we have added more description/explanation together with a more straightforward scheme to further clarify the concept of 3D OSS (to avoid the confusion with multiple molecular layers) in the revised manuscript as follows:

Page 3: *"As an efficient bottom-up approach, on-surface synthesis (OSS) defines a special opportunity to investigate intermolecular coupling at sub-molecular level and has successfully delivered various covalently-*

bonded two-dimensional (2D) polymers [6–12]. So far, these systems were all based on the lateral coupling of molecular precursors within a single molecular layer. *To extend OSS from two to three dimensions, it requires not only the lateral coupling parallel to the substrate but also the covalent bonding perpendicular to the substrate, e.g., interlayer coupling between molecular layers, which is yet to be realized.*

Page 3: *“Especially, the unique 3D conjugated electronic structure and the geometric combination of pentagons and hexagons of C_{60} molecules make their reactions significantly distinct from those of graphene or planar polyaromatic molecules [23,24]. The electron-deficient curved π -conjugation allows C_{60} to form adducts with various dienes (e.g. cyclopentadiene, furan, anthracene) as a dienophile [23,25]; due to the non-planar multiple reactive sites ([6,6] bonds), C_{60} is expected to act as a steric spherical “multi-plug” reactant [20,26,27].”*

Page 4: *“As demonstrated by scanning tunneling microscopy (STM) imaging and density functional theory (DFT) calculations, interlayer [4+2] cycloaddition between C_{60} and BCPM perpendicular to the substrate is triggered upon thermal activation. The resultant adduct, i.e. C_{60} -BCPM, shows obvious difference from pristine C_{60} , including the frozen orientation hence triangular sub-molecular feature observed at room temperature (RT), increased adsorption height, distinct domain structure and reactivity. Especially, forming macrocycles with the hexamer rings with C_{6v} symmetry dominated, lateral intermolecular [2+2] cycloaddition among C_{60} -BCPMs parallel to the substrate is evidenced. Using C_{60} and BCPM as the first example, this work presents an unconventional strategy for bottom-up synthesis of 3D covalently-bonded organic architectures, extending OSS from 2D to 3D by cycloaddition reactions with fullerenes.”*

Page 12: *“As demonstrated by our experimental results and DFT calculations, thanks to the multiple reactive sites of C_{60} , both the vertical covalent bonding via interlayer [4+2] cycloaddition between C_{60} and the phenyl ring of BCPM and the lateral covalent bonding between the C_{60} heads of C_{60} -BCPMs via [2+2] cycloaddition are constructed, representing a prototype of 3D synthesis on the surface.”*

Page 13: *“Allowing both vertical and lateral covalent bonding, this work unlocks an efficient route for bottom-up synthesis of covalently-bonded 3D organic framework on surfaces via cycloaddition of organic compounds with fullerenes--the steric multi-plug reactants, thus extending OSS from 2D to 3D.”*

Fig. 1 | Schematic illustration for the cycloaddition of C₆₀ and BCPM on Au(111). 2D OSS is based on the dissociation/formation of organic precursors' σ bonds at the planar active sites of precursors, allowing only lateral covalent bonding within a single molecular layer thus confining the synthesis to be 2D. In addition to the lateral covalent bonding, to extend OSS from 2D to 3D, covalent bonding perpendicular to the surface is also required. In this work, BCPM molecules (C₁₂H₇NO₆), maleimide derivative equipped with *bis*(carboxylic acid)-phenyl group, assemble into 2D ordered honeycomb network on Au(111), where carbon, nitrogen, oxygen, hydrogen atoms are in grey, blue, red, white, respectively. C₆₀ layer is constructed on the BCPM layer. One C₆₀ sits on atop of the phenyl ring of one BCPM, providing appropriate steric configuration for the coupling between the LUMO of C₆₀ and the HOMO of BCPM. Upon thermal activation, **vertical** [4+2] cycloaddition between the phenyl ring of BCPM and [6,6] bond of C₆₀ is triggered. Thanks to the multiple reactive sites of C₆₀, the resultant C₆₀-BCPM molecules can laterally bond with one another by [2+2] cycloaddition between [6,6] bonds of their C₆₀ heads. In this way, both lateral and vertical covalent bonding is realized, representing a prototype for 3D synthesis on surfaces.

2. As it is claimed in the manuscript (p5, top) the underlying surface and the molecule-molecule play a role in the formation of the BCPM network. Then, the growth of the network of fullerenes is also driven by the underlying BCPM layer (p7, top). Due to these claims, the extension toward 3D is not straightforward and it cannot be simulated by only three molecules in the vacuum.

Answer: It is correct that, for on-surface synthesis, the templating of surface (molecule-substrate interaction) and intermolecular interaction both play an important role for the initial assembly of the molecules and on-surface reactions; in the present system, C₆₀ sitting on top of the phenyl ring of BCPM provides appropriate steric configuration for the coupling between the LUMO of C₆₀ and the HOMO of BCPM. However, the 3D OSS we described in the manuscript is straightforward and well supported by both our experimental results and DFT

modeling (Figs. 4–6). We have clearly demonstrated that, thanks to this “C₆₀ on BCPM” configuration and the multiple reaction sites of C₆₀, both lateral and vertical covalent bonding is realized, representing a prototype for 3D synthesis on surfaces.

3. To sum up, I'm still not convinced by this answer. After a very fine (re-)reading of this article, the transition to 3D is still only a hypothesis based on numerical simulations (without underlying surface, without self-assemblies) and the use of the conditional mode by the authors confirms this point of view. If the main scientific issue of this article is the extension towards the 3rd dimension (as indicated in the title and the introduction) then I think that the authors should give an experimental proof of this 3D extension.

Answer: Please refer to our responses to Comments #1.

Reviewer: # 2

Comments:

The authors have responded all the comments I gave with resonable reply, thus I would recomanded this paper to be published in Nature Communication.

Answer: We are grateful to the reviewer for taking the time to evaluate our revision and recommending acceptance of our revised manuscript.

Reviewer: # 3

Comments:

The authors made significant changes to the manuscript, in which they addressed most of my concerns and of the other reviewers. The revised version of the manuscript is clearer and scientifically more rigorous. Since I'm not an expert in the field of on-surface synthesis I cannot evaluate the immediate impact or application of this paper in the respective scientific community. However, in my opinion, this work is a beautiful example of molecular and supramolecular chemistry at the surface, and merits publication in Nature Communications.

The experimental methodology was well-planned, and the results seem robust and consistent. The sequential experimental characterization of the surface is for me the strongest point of this work.

Answer: We are grateful to the reviewer for taking the time to evaluate our revision and raising the positive remarks on our revised manuscript. Indeed, the constructive advices/comments raised by this reviewer in the last round of evaluation have helped us improve the article substantially.

I'm still a bit skeptical about the computational results. My biggest concern is that according to the DFT results, there is no significant driving force for the [4+2] cycloaddition reaction (it is highly endothermic and, in principle, not favorable in entropic terms either). However, it appears to occur in appreciable extent at the Au surface and be rather stable, considering the low reverse barrier of about 20 kJ/mol. There is my general opinion that DFT is not that accurate for relatively complex systems. There are many electronic effects that, being erroneously accounted for by the computational method can easily scale-up to significant errors. Many times, obtaining accurate computational energies requires a careful balance of effects (like for instance, the use of homodesmic schemes for canceling effects in reactants and products). Given this, in my opinion, the interpretation of the experimental STM results is excessively anchored on the DFT results. I would appreciate more experimental work to validate some of the claims, for example (a) TGA of a solid C₆₀-BCPM sample to evaluate decomposition temperature; (b) characterization of the hexamers after desorption of the surface (e.g. by washing). I understand, however, that carrying out these studies is not trivial and that the paper already reports a good amount of valid experimental data to justify publication as a communication.

Answer: We do enjoy the communication/discussion with this reviewer. About the “driving force” for [4+2] cycloaddition between C₆₀ and BCPM, we have the following explanation. [4+2] cycloaddition is known as a reversible reaction; upon annealing, the reactions in both directions (the forward and reverse reactions) will both happen when a thermal equilibrium is established, with the population of the initial and final states to be characterized by the Boltzmann’s distribution associated with the energies of the two states. When the annealing stops, the metastable states resulted after the cycloaddition can be kinetically trapped. Similar cycloaddition results have been demonstrated in the earlier experiments [e.g. *Chem. Eur. J.* **25**, 9902–9912 (2019)].

For the suggested TGA experiments, we have addressed this question to Reviewer #2 in the last round of evaluation and Reviewer #2 believes it is “reasonable reply” without further questions. Our response was as follows: “As C₆₀-BCPM is fabricated using on-surface synthesis (OSS), it is rather challenging to extract it from Au(111) as well as to obtain a sufficient amount of C₆₀-BCPM for TGA. This is a common issue for almost all products obtained from OSS.” In fact, as demonstrated in Fig. 5, the cycloaddition between C₆₀ and BCPM forming C₆₀-BCPM occurs at 370 K, and survive the annealing at 490 K for further cycloaddition (Fig. 6), indicating that the [4+2] cycloaddition between C₆₀ and BCPM is stable even at a much higher annealing temperature. For the suggestion of “characterization of the hexamers after desorption of the surface (e.g. by washing)”, it is experimentally unattainable: as all these experiments are performed in an ultrahigh vacuum (UHV) chamber, it is not possible to wash the surface by solvent; also, introducing solvent or taking the sample out of UHV will contaminate the sample and compromise the results; once the hexamers are desorbed from the surface, they are soon pumped away in UHV.

Some minor corrections:

- Page 3, correct “comound ” to “compound ” in “...magnetism of TDAE-C60 comound [19] ”

Answer: We thank the reviewer and have made the correction accordingly.

- Page 12, top, where it says “C60-BCPM is also lower than that of C60 (Supplementary Fig. 12) ”, referring to the calculated energy barrier to [2+2] cycloaddition; according to Supplementary Fig. 12 this barrier is lower to C60 (1.77 eV compared to 1.93 eV to C60-BCPMs). Hence, the following conclusion “In this regard, [2+2] cycloaddition among C60 parts of C60-BCPMs is promoted compared with the case of pristine C60 ” does not apply. Were the authors referring to the energy of the reaction, instead of the energy barrier? Please correct.

Answer: We are grateful to the reviewer for pointing out this error. We have made the correction in Supplementary Fig. 12.

Reviewers' Comments:

Reviewer #3:

Remarks to the Author:

In this revision, the authors have improved the manuscript to the point that I consider it suitable for publication in Nature Communications.

Reviewer: #3

Comments:

In this revision, the authors have improved the manuscript to the point that I consider it suitable for publication in Nature Communications.

Answer: We are grateful to the reviewer for taking the time to re-evaluate our revision and recommending acceptance of our article in Nature Communications.